# VSCoDe: Visual-Augmentation Selection for Contrastive Decoding

**Sihyeon Kim***  *sihk@kaist.ac.kr*
*KAIST AI*

**Boryeong Cho***  *venntum@kaist.ac.kr*
*KAIST AI*

**Sangmin Bae**  *bsmn0223@kaist.ac.kr*
*KAIST AI*

**Sumyeong Ahn†**  *sumyeongahn@kentech.ac.kr*
*KENTECH*

**Se-Young Yun†**  *yunseyoung@kaist.ac.kr*
*KAIST AI*

**Reviewed on OpenReview:** *https: // openreview. net/ forum? id= CqSyPc9W7Y*

## Abstract

Despite the impressive performance of recent Large Vision-Language Models (LVLMs), these models often produce inaccurate responses. To address this issue, previous studies have aimed to reduce hallucinations by using contrastive decoding (CD) with modified images, such as cropping objects related to query or adding noise, thereby contrasting with the original image. However, these methods have several limitations. First, employing fixed visual augmentation, such as adding noise, is a simple approach but too rigid to contrast on various queries. Conversely, using semantics in queries or images by leveraging external models can adaptively generate contrastive images, but it entails significant additional costs. To address these shortcomings, we explore using pre-defined visual augmentations to enable flexible adaptation to each query without relying on external models. We observe that each query achieves different contrasts through different visual augmentations. Based on this, we propose a novel method called VSCoDe, **V**isual-augmentation **S**election for **Co**ntrastive **De**coding, which adaptively selects augmentations using a proposed distance metric to identify those with higher contrast. Our empirical evaluations demonstrate that VSCoDe outperforms previous methods and enhances the quality of various vision-language tasks without additional training or reliance on external models.

## 1 Introduction

Pre-trained Large Vision-Language Models (LVLMs) (Liu et al., 2024a; Ye et al., 2023; Zhu et al., 2023; Dai et al., 2024; Li et al., 2022; 2023a; Radford et al., 2021) have gained prominence due to their ability to understand multiple data formats, especially vision and language, simultaneously. These models have demonstrated exceptional performance in various tasks such as zero-shot image classification (Radford et al., 2021; Yao et al., 2021), image-text retrieval (Yao et al., 2021; Li et al., 2022), visual question answering (Dai et al., 2024; Liu et al., 2024a), and image captioning (Li et al., 2022; 2023a). Unlike earlier encoder-based models like CLIP (Radford et al., 2021), most recent large-scale VLMs, such as LLaVA (Liu et al., 2024a),

---

* Authors contribute equally
† Corresponding Authors

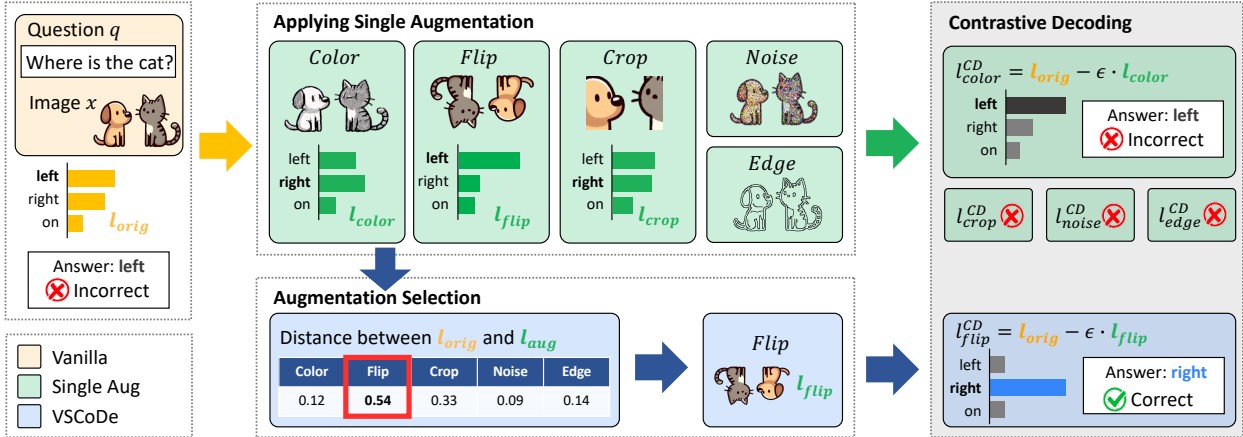

Figure 1: CD with a single augmentation may not yield correct answers for all types of questions, as it may fail to modify the key visual feature related to the semantics of the query. For example, the visual feature of the position-related query is affected by flip augmentation, whereas color augmentation does not. Therefore, it is essential to apply the appropriate augmentation for each query when applying CD. VSCoDe selects the augmentation with the largest distance $D(\cdot)$ we define to identify the augmentation that modifies the key feature of the image relative to the question. VSCoDe successfully selects the augmentation, enabling CD to produce the correct answer.

MPlugOWL (Ye et al., 2023), MiniGPT-4 (Zhu et al., 2023), and InstructBLIP (Dai et al., 2024), utilize autoregressive language decoders to expand their functionality, allowing them to cover more complex tasks.

However, language decoders sometimes produce incorrect outputs, a phenomenon called hallucination. Among various methodologies (Wei et al., 2022; Rose et al., 2023; Shao et al., 2024), a promising approach is contrastive decoding (CD) (Li et al., 2023b), which generates final answers by examining original outputs through contrastive outputs. More precisely, CD works in two stages: (1) generating output distributions given original and contrastive prompts each, and (2) subtracting the two output distributions to reduce the likelihood of hallucinated tokens. The effectiveness of CD relies on how well the contrastive prompt is generated and facilitates the contrastive predictions effectively. While creating contrast in language models is relatively straightforward –by replacing original words with their opposites or random words (Kim et al.; Wang et al., 2024)– vision-language models require a more deliberate approach as no clearly defined strategy exists for generating contrastive images.

Several works have been performed on the generation of contrastive images, but they are limited to various semantics considerations. VCD (Leng et al., 2023) adds Gaussian noise to the image, but does not consider the semantics of the query. On the other hand, some studies, such as HALC (Chen et al., 2024) and CRG (Wan et al., 2024), have attempted to understand semantics to generate object-manipulated images. However, they are limited to object-level semantics and involve the additional cost of using external models for object detection. Therefore, we pose the question, *"How can we leverage pre-defined cheap visual augmentation operations while incorporating semantic understanding?"*

To answer our question, we first empirically observe that the augmentation required to generate contrast varies depending on the query's semantics. For instance, as shown in Figure 1, when the question pertains to position, using position-related augmentations such as Flip creates significant contrast. Consequently, selecting augmentations that induce more contrastive predictions for each query can make CD more effective. Building on this intuition, we hypothesize that augmentations producing greater distances between the logits of the original and augmented images facilitate contrasting predictions.

Based on our findings, we propose VSCoDe, a novel method for automatically selecting augmentation for CD, which generates appropriate augmentation based on the semantic of query without requiring additional training or external model. As illustrated in Figure 1, VSCoDe involves three steps: (1) given multiple candidate augmentations, provide various types of augmented images to VLMs and generate a single token output on

each image, (2) measure the distance $D(\cdot)$ between output distributions of the original and augmented images, and (3) select the most contrasting output with the maximum distance $D(\cdot)$ to achieve the final CD result.

Our contributions can be summarized as follows:

- We explore the effect of visual augmentation on LVLMs. Our findings indicate that each augmentation has a distinct impact on the given question, altering the output distribution of VLMs and subsequently affecting the response. It highlights the importance of using appropriate augmentation depending on the query.

- Based on the findings, we introduce an algorithm called `VSCoDe` that selects contrastive augmentation to empower CD capability without additional training or using external models. Through distance $D(\cdot)$, `VSCoDe` automatically determines the appropriate visual augmentation for given query semantics, thereby achieving a higher level of CD effect.

- We extensively evaluate the performance of `VSCoDe` across various tasks, including Visual Question Answering (VQA) on the MME (Fu et al., 2024), MMBench (Liu et al., 2024b), VQAv2 (Goyal et al., 2017), and POPE (Li et al., 2023c), as well as captioning tasks on the MSCOCO (Lin et al., 2015) dataset using LVLMs . `VSCoDe` shows improved performance on each task, and in particular, on the MME benchmark with the LLaVA-1.5 7B model, `VSCoDe` outperforms both single augmentation approaches and previous CD-based methods, yielding a $1.82\times$ larger performance gain compared to previous methods.

## 2 Preliminaries

Here, we provide a concise summary of background information to aid in understanding this research. We specifically discuss LVLMs, visual data augmentation, and contrastive decoding.

**Generative LVLMs.** LVLMs are among the most prominent multi-modality models. They process pairs of input image $v$ and text (*e.g.,*question) $q$, denoted as $(v, q)$, and generate answers by utilizing the visual information within $v$. In this paper, we primarily focus on generative LVLMs, rather than CLIP-like (Radford et al., 2021) models. These generative LVLMs produce tokens one at a time in sequence similar to LLMs. The mathematical expression for this process is:

$$y_t \sim p(y_t|v, q, y_{<t}).$$

Here, $p(\cdot)$ represents the softmax of the output of the vocabulary set, and $y_{<t}$ denotes the tokens generated up to but not including the timestamp $t$. Like LLMs, LVLMs are also prone to hallucination (Li et al., 2023c; Liu et al., 2023a; Tong et al., 2024), where the model erroneously assigns higher probabilities to tokens that do not factually exist in the provided image.

**Visual Augmentation (VA).** VA consists of long-established techniques that modify visual data to produce desired images for computer vision research, such as enhancing sharpness, adjusting color jitters, and more. While some augmentation techniques, like mixup (Zhang et al., 2017) or CutMix (Yun et al., 2019), require combining more than two images, our discussion focuses on single-image augmentation operations for simplicity. We focus on a specific framework:

$$v' = \mathcal{O}_{o\in\mathcal{A}}(v),$$

where $\mathcal{O}(v)$ represents a manipulation on the image $v$, and $o$ an augmentation within the set $\mathcal{A}$. In this paper, we employ the augmentations $\mathcal{A} =\{\texttt{color}, \texttt{flip}, \texttt{random crop}, \texttt{random erase}, \texttt{sharp}, \texttt{edge}, \texttt{noise}\}$. Examples are illustrated in Figure 2. The descriptions of the augmentations are: (1) color: color inversion, (2) flip: horizontal flip followed by vertical flip, (3) crop: cropping a random part of the image, (4) erase: randomly erasing part of the image, (5) sharp: adjusting image sharpness, (6) edge: extracting edge textures, and (7) noise: adding diffusion noise. Note that we use the default noise setting from VCD (Leng et al., 2023).

**Contrastive Decoding (CD).** The CD approach was first introduced in the language domain (Li et al., 2023b). It operates by generating two outputs using two different models: an expert model that produces the original outputs and an amateur model that generates contrastive outputs. The CD is then performed based on the contradictions between them. This idea has also been explored in the vision-language domain,

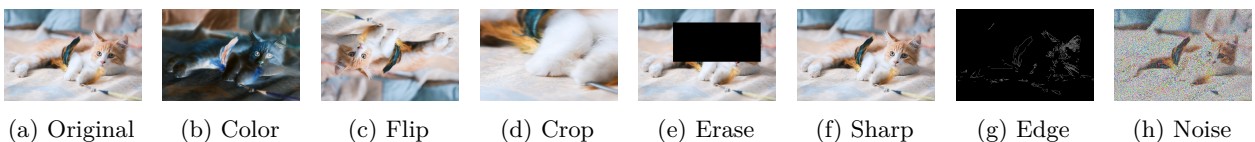

(a) Original    (b) Color    (c) Flip    (d) Crop    (e) Erase    (f) Sharp    (g) Edge    (h) Noise

Figure 2: Examples of visual augmentations utilized in this paper.

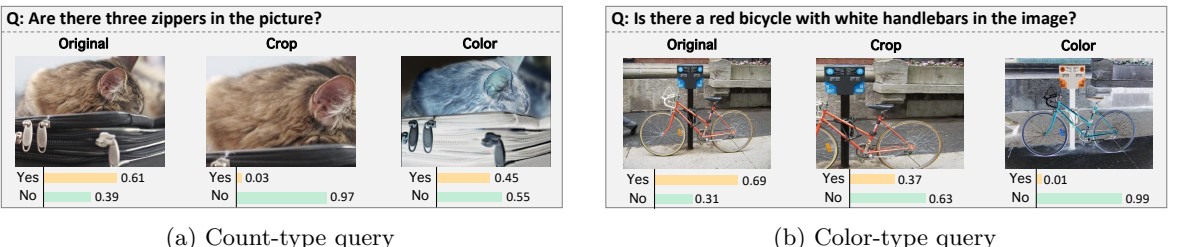

(a) Count-type query          (b) Color-type query

Figure 3: The examples show output probabilities from the MME benchmark using the LLaVA-1.5 7B model. The influence of augmentation varies depending on the query type. For instance, in (a) with a count-type question, a cropped image generates a contrastive distribution, whereas a color-changed image does not. Conversely, in (b), a color-related question is affected by color augmentation.

where manipulated images are used to induce contrastiveness and help remove unrelated information, such as hallucinations. Instead of employing two separate models, the VLM setting uses original and manipulated images as two distinct inputs to make the contrast. The process operates as follows:

$$p_{\text{CD}}(y|v, q, \mathcal{O}) = (1 + \alpha)\texttt{SOFTMAX}\Big(f(y|v, q)\Big) - \alpha\texttt{SOFTMAX}\Big(f(y|\mathcal{O}(v), q)\Big), \tag{1}$$

where $f(\cdot)$ is the output logit obtained from VLM, and $\alpha$ as hyperparameter for the strength of contrastiveness. To amplify the inherent hallucinations in VLMs, VCD (Leng et al., 2023) introduced noise to the image, HALC (Chen et al., 2024) applied cropping, and CRG (Wan et al., 2024) employed object-wise erasing. Another approach, ICD (Wang et al., 2024), involves modifying the text prompt to create confusion in the model's outputs. Subsequently, they subtracted the logits of the hallucinated image from the logits of the original image.

## 3   VSCoDe: Visual-Augmentation Selection for Contrastive Decoding

This section explores the impact of VAs on LVLMs, focusing specifically on CD. In essence, we demonstrate that certain VAs cause either contrast or persistence on key visual feature, implying that the output distribution of the augmented image either varies or stays consistent with the distribution of the original image for the given query. Furthermore, we detail our discovery that contrastive augmentation can be identified using the distance metric we propose, which relies on the difference between probability distributions. Building on these insights, we present a novel algorithm named VSCoDe, which selects contrastive augmentation among multiple candidate augmentations for the given query.

### 3.1   Query-Dependent Augmentation Effect

We begin by demonstrating that augmentations affect LVLM outputs differently, depending on the information required by the query from the image. This difference occurs because an augmentation that modifies a crucial feature needed for the query results in a shift in the LVLM's output. Conversely, when the essential feature is left unchanged, the output remains stable. We define these cases as contrastive augmentation and persistent augmentation, respectively. Figure 3 illustrates the output distribution from differently augmented images, where the bar represents probabilities computed with the softmax function of logits. In the first example, the *crop* augmentation is a contrastive augmentation, altering the key feature for a count-related question, leading to a significant change in probability distribution. Conversely, *color* augmentation is classified as

Table 1: The values indicate the net change from CD per category, with blue and yellow marking the best and worst results. No augmentation performs best across all categories, and effectiveness varies. Results are averaged over five seeds on the MME benchmark using LLaVA-1.5 7B.

| | **Category** | | | | | | | | | |
|---|---|---|---|---|---|---|---|---|---|---|
| **Aug** | Ext | Cnt | Pos | Clr | Pst | Cel | Scn | Lmk | Art | OCR |
| Noise | -2 | 20 | 2 | -15 | -6 | 69 | 41 | 69 | 37 | -2 |
| Color | -2 | 11 | -6 | 10 | 60 | 76 | 42 | 82 | 68 | -15 |
| Crop | 2 | 13 | -4 | -11 | 33 | 32 | 49 | 15 | 11 | -14 |
| Edge | -4 | 18 | 0 | 0 | 68 | 91 | 58 | 93 | 27 | -13 |
| Flip | -1 | 8 | 5 | 2 | 36 | 35 | 31 | 53 | 10 | -5 |
| Erase | 1 | 12 | -11 | -4 | 4 | 0 | 38 | 26 | 26 | -14 |
| Sharpness | 3 | 2 | -5 | -2 | 13 | -24 | 41 | 3 | 14 | -17 |

Figure 4: Both (a) and (b) show `Gain` measured on the MME benchmark using all augmentations in $\mathcal{A}$. (a) Larger distance $D(\cdot)$ yields larger `Gain`. (b) Choosing augmentation with the largest $D(\cdot)$ is most effective in achieving high `Gain`.

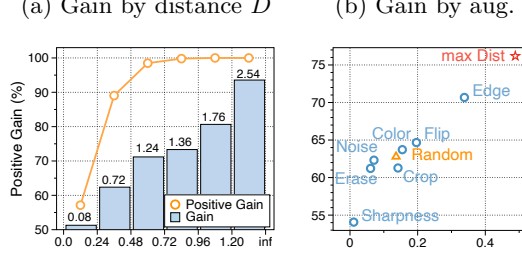

(a) Gain by distance $D$     (b) Gain by aug.

persistent augmentation, having a similar probability distribution. This effect is similarly observed in the second example.

This effect results in varied performance on the LVLM. To demonstrate the distinct impact of each augmentation on different queries, we evaluated CD performance with each augmentation applied to the MME benchmark. The perception tasks in the MME benchmark are categorized into 10 groups, including color, count, and existence. Observing performance in each category enables us to clearly assess the impact of each augmentation on specific question types. Table 1 shows the difference between the number of samples corrected by CD and those that became incorrect due to CD across categories for each augmentation. Positive values, especially higher ones, indicate effective augmentations that likely act contrastively for the respective query type, whereas negative values suggest that an augmentation may not be suitable for that query type. For instance, changing the color of the image acts contrastively for color-related queries by leading to incorrect predictions, while it remains consistent with position-related queries, as positional information is unchanged. This underscores that no single augmentation is universally effective, emphasizing the importance of selecting the optimal augmentation for each question.

For effective CD, contrastive augmentation should be applied to guide the LVLM toward generating incorrect answers, thereby increasing the logit for the correct token while decreasing the logit for the wrong token after logit subtraction. As the contrastiveness of each augmentation depends on the visual information required by each question, selecting an appropriate augmentation is crucial. To address this, we propose an automatic augmentation selection strategy that identifies the optimal contrastive augmentation for each question.

## 3.2 Maximizing Contrast: Selecting Visual Augmentation with the Largest Distance

The remaining challenge is to select a suitable visual augmentation that yields the most contrastive outputs relative to inference on the original image. In this section, we outline the indicators for effective contrastive augmentation. Intuitively, applying CD should lead to (1) an increase in logit for the correct token and (2) a decrease for the incorrect token. Based on the intuition, we hypothesize that the augmentation resulting in the most different output distribution can serve as a contrastive augmentation. To measure the distribution difference $D(\cdot)$ between the output distributions from original and augmented images, we use $L_2$-norm.

$$D(v, q, \mathcal{O}) = \left\| \left( \texttt{SOFTMAX}(f(y|v, q)) - \texttt{SOFTMAX}(f(y|\mathcal{O}(v), q)) \right) \right\|_2 . \tag{2}$$

Note that we mainly use $L_2$-norm hereinafter, and report analysis of various metrics in Appendix A.

To assess the effectiveness of CD, we use the logit increase of the ground truth token achieved through CD as a key indicator. This measure, referred to as `Gain`, is calculated as follows:

$$\texttt{Gain}(v, q, \mathcal{O}) = p(y_{\text{GT}}|\mathcal{O}(v), q) - p(y_{\text{GT}}|\mathcal{O}_{original}(v), q). \tag{3}$$

where $y_{\text{GT}}$ is the ground truth token and $\mathcal{O}_{original}$ is a transformation that returns the original image. To verify our hypothesis – *a bigger distance $D(\cdot)$ can have the most contrastiveness* – we measure the Gain and $D(\cdot)$ on the MME benchmark using augmentations from $\mathcal{A}$ across 5 seed runs. As shown in Figure 4a, as $D(\cdot)$ increases, the Gain gets bigger, most resulting in positive Gain values, which supports our hypothesis. Thus, choosing the augmentation with the largest $D(\cdot)$ can be an effective strategy for maximizing the Gain. To test this approach, we select the augmentation that has the biggest $D(\cdot)$ on each augmentation and average their Gain scores and percentage of positive Gain scores. As shown in Figure 4b, we confirm that the augmentation with the greatest $D(\cdot)$ yields the highest average increase in the Gain score. Notably, this increase surpasses that of any single augmentation, validating the effectiveness of a $D(\cdot)$-based augmentation selection strategy. This finding supports using the augmentation with the largest $D(\cdot)$ as the contrastive augmentation $o$ for each query $q$.

---

**Algorithm 1** VSCoDe: Visual-Augmented Contrastive Decoding

---

**Input**: Image and question pair $(v, q)$, target sequence length $T$, Augmentation set $\mathcal{A}$, The number of of Visual augmentations $N$, Distance function $D(\cdot)$, Amplification coefficient $\alpha$ for CD, plausibility constraint parameter $\beta$

**for** $t = 1...T$ **do**

    **if** $t = 1$ **then**                          ▷ Determine contrast augmentation for the entire decoding process

         $z_t \leftarrow f(y_t|v, q, y_{<t})$ and $\tilde{z}_{t,i} \leftarrow f(y_t|\mathcal{O}_o(v), q, y_{<t}), \quad \forall o \in \mathcal{A}$        ▷ Generate logits

         $p_t \leftarrow \text{SOFTMAX}(z_t)$ and $\tilde{p}_{t,i} \leftarrow \text{SOFTMAX}(\tilde{z}_{t,i}), \quad \forall o \in \mathcal{A}$        ▷ Compute probability

         $\hat{o} \leftarrow \arg\max_{o \in \mathcal{A}}(D(p_t, \tilde{p}_{t,i}))$                  ▷ Select the most constrastive augmentation

    **else**

         $z_t \leftarrow f(y_t|v, q, y_{<t})$ and $\tilde{z}_{t,\hat{o}} \leftarrow f(y_t|\mathcal{O}_{\hat{o}}(v), q, y_{<t})$        ▷ Generate logits

         $p_t \leftarrow \text{SOFTMAX}(z_t)$ and $\tilde{p}_{t,\hat{o}} \leftarrow \text{SOFTMAX}(\tilde{z}_{t,\hat{o}})$        ▷ Compute probability

    **end if**

     $p_{\text{VSCoDe},t} = (1 + \alpha) \cdot p_t - \alpha \cdot \tilde{p}_{t,\hat{o}}$        ▷ Compute VSCoDe probability

     $V_{\text{cand}}(y_{<t}) \leftarrow \{y_t \in V : p_t(y_t|v, q, y_{<t}) \geq \beta \max_w p_t(w|v, q, y_{<t})\}$        ▷ Candidate Set

     $p_{\text{VSCoDe},t}(y) = 0$, if $y \notin V_{\text{cand}}(y_{<t})$        ▷ Discard not-candidate words

     $y_t = \text{SAMPLING}_y(p_{\text{VSCoDe},t})$        ▷ Sampling next word

**end for**

---

### 3.3 Proposed Algorithm

Based on the above observations, we propose VSCoDe to automatically select an appropriate augmentation for each query by utilizing the $D(\cdot)$. The entire procedure is summarized in Algorithm 1. In the initial decoding phase with the given question, we adaptively select contrastive augmentation by choosing the augmentation with the maximum $D(\cdot)$. This produces a similar effect to applying CD for every token while significantly reducing the computational cost. The chosen augmentation $\hat{o}$ is then used for the remainder of the decoding process. Once the contrastive augmentation is determined, LVLM calculates the probability of tokens $p_{\text{VSCoDe},t}$ using Eq. (1). Subsequently, among the whole vocabulary $V$, the candidate vocabulary set $V_{\text{cand}} \in V$ is defined to select a more reliable token following the original CD algorithm (Li et al., 2023b). This process is repeated iteratively to generate the output words $y_t$.

We use two scenarios to define candidate augmentations: *All* and *Coreset*. *All* uses all the augmentations in the candidate set $\mathcal{A}$. However, due to the characteristics of data distributions, certain augmentations may not yield meaningful contrast effects and may be effectively replaced with alternative ones. In this case, excluding these augmentations may work to eliminate noisy augmentations. So we introduce the *Coreset* strategy that leverages validation to choose a subset of augmentations $\mathcal{A}' \subset \mathcal{A}$ to use only the more effective augmentations.

### 3.4 Coreset Strategy

**Removing noisy augmentations via acceptance threshold $\tau$.** Using the $D(\cdot)$, we aim to select a VA that shows high-performance improvement when applied to CD. However, there may exist cases where certain VAs may not be appropriate contrastive augmentation for a specific task. In this case, these VAs contribute

less to performance improvement than other VAs on average and can sometimes become noise that prevents other VAs from being chosen as contrast. To determine these, We introduce the acceptance threshold $\tau$, a simple baseline that eliminates the noise VAs. To determine the suitability of VAs for the target task, we utilize the LVLM's first token generation distance by `VSCoDe` for each VA in the sample sub-dataset. Let $c_i$ be the number of times that $VA_i$ selected as contrast VA among a total of $M$ VAs. For the $N$ data samples and acceptance threshold $\tau$, candidate VAs with $c_i < \tau \frac{N}{M}$ are treated as unsuitable for this task and removed. We used the $\tau = 0.5$ as a baseline throughout the main experiments.

## 4 Experiments

### 4.1 Experimental Settings

**Datasets and evaluation metrics.** We conduct experiments on Visual Question Answering (VQA) tasks and captioning tasks. VQA task evaluates how well LVLM generates robust and correct answers to various questions, and the captioning task measures how well the LVLM generates captions given the image. We use MME (Fu et al., 2024), MMBench (Liu et al., 2024b), VQAv2 (Goyal et al., 2017), and POPE (Li et al., 2023c) benchmarks for the VQA task, and each dataset consists of image-question pairs. For the captioning task, we evaluate on MSCOCO (Lin et al., 2015).

- **MME** is an LVLM evaluation dataset with granular question categories, including 10 categories from the perception tasks and 4 from the cognition tasks. The labels consist of 'Yes' or 'No,' and performance is measured by MME score, which is derived from accuracy. In this paper, we evaluate the perception tasks as our method focuses on observation ability.

- **MMBench** is a dataset of image-question pairs from 20 categories to validate how skillfully LVLM performs on various vision-language tasks with given option labels. For evaluation, we incorporate CircularEval, which rotates the positions of the possible option labels in a circular manner.

- **VQAv2** is a dataset containing open-ended questions paired with images. This allows for a proper evaluation of how expertly the model can utilize the given visual information rather than simply using the learned language priors of the decoder. We randomly select 30K samples from the VQAv2 evaluation dataset to validate our method.

- **POPE** benchmark is proposed to evaluate the hallucination of LVLMs. It asked whether the given object exists in the image or not. The fake object is generated from three sampling strategies: random, popular, and adversarial. Images are from three sources: MSCOCO (Lin et al., 2015), A-OKVQA (Schwenk et al., 2022), and GQA (Hudson & Manning, 2019), and we report the average performance across the sampling strategies on each image source.

- **MSCOCO** dataset provides 328K images for various tasks, such as object detection, or captioning. We use 500 random images from the validation set to generate image captions and evaluate with CHAIR (Rohrbach et al., 2018) metric, which is designed to evaluate hallucination in image captioning models. It assesses how accurately generated captions reflect the visual content of an image without including irrelevant details. CHAIR_{I,S} metrics each assess on alignment with the image and semantic plausibility.

For the reliability of the results, we report performance using the average of the results of 5 different seed runs for MME and MMBench, and a single run for VQAv2.

**Models.** We evaluate the performance of `VSCoDe` using state-of-the-art LVLMs, mainly on LLaVA-1.5 (Liu et al., 2023b). Specifically, we use pre-trained LLaVA-1.5 with Vicuna (Chiang et al., 2023) 7B and 13B as the language decoder. Additional experiments on various LVLMs are provided in Appendix B.

**CD methods.** We use 7 augmentations in Figure 2 set $\mathcal{A}$. For the baseline, we use vanilla setting that do not use CD, and CD with each single augmentation. To evaluate the effectiveness of `VSCoDe`, we compare with VCD (Leng et al., 2023), ICD (Wang et al., 2024), CRG (Wan et al., 2024) and HALC (Chen et al., 2024). Note that a single noise addition augmentation is equivalent method to the VCD. For `VSCoDe`, we

Table 2: MME score on perception tasks using the LLaVA-1.5 7B. *Q-Dep* denotes that the method varies the contrast based on the given query, and *Ext* specifies whether an additional external model is employed. *Select* denotes that the choice of augmentations is selected by a *Metric*, instead of a fixed single augmentation. Note that *All* uses a set of seven augmentation types, including noise, but excluding text augmentation, while *Coreset* utilizes a subset of four augmentations (Color, Edge, Crop, Flip) selected by the acceptance threshold $\tau$. The best and second-best performances are reported using **bold** and underline formatting, respectively.

| Method | Type | Q-Dep | Ext | Select | Metric | Existence | Count | Position | Color | Posters | Celebrity | Scene | Landmark | Artwork | OCR | Total |
|---|---|---|---|---|---|---|---|---|---|---|---|---|---|---|---|---|
| Vanilla | - | - | - | - | - | 180.00 | 112.00 | 117.67 | 147.00 | 121.02 | 110.24 | 148.80 | 129.15 | 109.35 | 97.00 | $1272.22_{\pm 28.28}$ |
| Single CD | Color | ✗ | ✗ | ✗ | - | 180.00 | 129.33 | 127.33 | 152.67 | 132.45 | 128.06 | 148.30 | 138.35 | 109.75 | 101.00 | $1347.24_{\pm 21.13}$ |
| | Edge | ✗ | ✗ | ✗ | - | 176.00 | 133.00 | 126.67 | 141.00 | 133.61 | 133.35 | 150.40 | 142.45 | 110.70 | 103.50 | $1350.68_{\pm 6.56}$ |
| | Sharp | ✗ | ✗ | ✗ | - | 184.00 | 113.33 | 127.00 | 163.67 | 126.46 | 111.59 | 150.80 | 132.15 | 112.60 | 102.00 | $1323.60_{\pm 6.51}$ |
| | Crop | ✗ | ✗ | ✗ | - | 183.00 | 122.00 | 129.67 | 151.67 | 129.86 | 128.35 | 146.80 | 134.70 | 110.95 | 101.50 | $1338.50_{\pm 23.83}$ |
| | Erase | ✗ | ✗ | ✗ | - | 182.00 | 118.33 | 119.33 | 153.67 | 126.12 | 115.24 | 149.95 | 134.85 | 112.90 | 98.50 | $1310.89_{\pm 18.82}$ |
| | Flip | ✗ | ✗ | ✗ | - | 178.00 | 124.67 | 129.67 | 152.33 | 132.24 | 125.24 | 147.30 | 137.05 | 111.25 | 107.00 | $1344.75_{\pm 21.05}$ |
| VCD | Noise | ✗ | ✗ | ✗ | - | 186.00 | 115.00 | 122.33 | 154.33 | 127.76 | 125.47 | 149.85 | 138.30 | 110.90 | 93.50 | $1323.44_{\pm 27.36}$ |
| ICD | Text | ✗ | ✗ | ✗ | - | 183.00 | 122.67 | 120.33 | 165.33 | 128.16 | 112.82 | 150.65 | 130.90 | 113.10 | 98.50 | $1325.47_{\pm 19.56}$ |
| CRG | Erase | ✓ | ✓ | ✗ | - | 185.00 | 116.00 | 115.33 | 125.67 | 144.49 | 132.88 | 158.40 | 135.95 | 101.80 | 94.50 | $1310.02_{\pm 20.99}$ |
| Random | *All* | ✗ | ✗ | ✓ | Rand | 182.00 | 125.67 | 127.33 | 153.00 | 130.82 | 123.94 | 148.90 | 134.80 | 110.80 | 102.50 | $1339.76_{\pm 32.19}$ |
| VSCoDe | *All* | ✓ | ✗ | ✓ | Dist | 179.00 | 137.33 | 129.67 | 146.67 | 133.95 | 131.88 | 147.40 | 142.00 | 112.50 | 108.50 | $\mathbf{1368.89}_{\pm 20.39}$ |
| | *Coreset* | ✓ | ✗ | ✓ | Dist | 179.00 | 138.33 | 129.67 | 150.67 | 132.31 | 130.88 | 145.00 | 139.75 | 110.25 | 108.50 | $\underline{1364.36}_{\pm 11.50}$ |

experiment on both *all* and *coreset* strategies. The augmentations used by *coreset* differs depending on the datasets. For example, on the MME benchmark, the LLaVA-7B model utilizes 4 augmentations: `color`, `edge`, `crop`, and `flip`. The implementation details for each method can be found in Appendix E.

**Implementation details.** We choose $\alpha = 1.0$ and $\beta = 0.1$ for the main experiment. Additionally, we use $T = 1.0$ and $p = 1.0$ for the sampling strategy, which employs the softmax distribution for the next token generation. Additionally, we conducted experiments under various decoding settings, and the corresponding ablation studies can be found in Appendix C.

## 4.2 Experiment Results

**Result on each category.** Table 2 presents the MME score of CD with various augmentations and different methods applied on the MME dataset across different perception categories, using the LLaVA-1.5 7B model. The baseline performance with individual augmentations shows an overall increase in the total MME score. This suggests that augmentations likely help create contrast in the visual features for CD. For example, in questions involving the identification of celebrities or landmarks, humans can still answer accurately even if the color of the image is altered. However, LVLMs may struggle to recognize objects when color, implying that color information is one of the important feature to recognize the object. Still, each augmentation is effective within specific categories, underscoring the need to choose suitable augmentations for optimal results.

Comparing with baselines, `VSCoDe` results the best performance. When using a single augmentation for CD of LVLMs, it is challenging to gain distinguished performance across all types of questions. In contrast, `VSCoDe`, which selects the most suitable visual augmentation based on the given question, demonstrates significant performance improvements across various question categories compared to single augmentation approaches. Furthermore, it surpasses both VCD and ICD, emphasizing the critical role of selecting appropriate augmentations. Note that we exclude HALC from the baseline as it fails to generate proper output on LLaVA-1.5 7B model. In Appendix D, we report the performance of LLaVA-1.5 13B and analysis on the selected augmentations on each categories by `VSCoDe`.

**Result on performance and cost comparison** Table 3 presents the evaluation scores for all datasets(MME, MMBench, VQAv2, POPE, BLEU, and CHAIR) used in the experiments on LLaVA-1.5 7B. Additionally, it includes the time and memory costs, which were measured during the captioning task. Although HALC records high performance in captioning task, it consumes big cost for using external model. In contrast,

Table 3: Performance on the various benchmarks and computational costs for different methods for LLaVA-1.5 7B. (M), (A), and (G) denote the MSCOCO, A-OKVQA, and GQA datasets for POPE. Detailed descriptions of each experiment are provided in Appendix E.

| Method | Augmentation | | | | | Score | | | | | | | | | Cost | |
| | Type | Q-Dep | Ext | Select | Metric | BLEU | CHAIR$_S$ | CHAIR$_I$ | POPE (M) | POPE (A) | POPE (G) | MME | MMBench | VQAv2 | Time | Mem |
|---|---|---|---|---|---|---|---|---|---|---|---|---|---|---|---|---|
| Vanilla | - | - | - | - | - | 40.13 | 8.00 | 5.47 | 83.63 | 80.44 | 80.85 | 1272.22 | 71.46 | 66.48 | 1.00 | 1.00 |
| VCD | Noise | ✗ | ✗ | ✗ | - | 47.98 | 6.40 | 4.24 | 83.40 | 81.41 | 81.53 | 1323.44 | 72.03 | 69.19 | 1.96 | 1.00 |
| ICD | Text | ✗ | ✗ | ✗ | - | 44.99 | 5.36 | 6.62 | 83.72 | 81.23 | 81.12 | 1325.47 | 72.05 | 70.27 | 1.97 | 1.00 |
| HALC | Segment | ✓ | ✓ | ✗ | - | 16.87 | 20.40 | 6.75 | - | - | - | 1114.16 | - | - | 54.27 | 1.45 |
| HALC$^\dagger$ | Segment | ✓ | ✓ | ✗ | - | 50.24 | 5.80 | 4.31 | 85.03 | - | - | - | - | - | 7.02 | 1.44 |
| CRG | Erase | ✓ | ✓ | ✗ | - | 42.70 | 5.20 | 3.37 | **85.28** | 82.27 | 82.25 | 1310.02 | 73.07 | 70.25 | 2.64 | 1.19 |
| Random | *All* | ✗ | ✗ | ✓ | Rand | 49.02 | 5.75 | 3.91 | 83.87 | 81.80 | 81.81 | 1339.76 | 72.49 | 69.41 | 1.96 | 1.00 |
| VSCoDe | *All* | ✓ | ✗ | ✓ | Dist | 51.37 | 4.80 | 3.21 | 84.4 | **83.06** | 82.75 | **1368.89** | **74.32** | **71.02** | 2.45 | 1.04 |
| | *Coreset* | ✓ | ✗ | ✓ | Dist | **51.97** | **4.60** | **3.10** | 84.65 | 82.75 | **82.84** | 1364.36 | 74.27 | 70.94 | 2.15 | 1.03 |

VSCoDe shows best performance with slight increase on the cost. Detailed setting for measuring the cost is described in Appendix E.

In the process of applying VSCoDe, our model requires inference as the number of VAs used in the first step only, and each subsequent generation step requires twice token generation stages, which is the same with other CD methods. Table 3 illustrates the cost on MSCOCO captioning task. They are denoted as a respective cost compared to the vanilla baseline, which does not utilize CD. In the case of HALC and CRG, since they use an external object detector, the amount of memory used and the time taken in the decoding process are longer. In particular, in the case of HALC, the basic reproduction process takes a long time because of generating very long interpretations. Although optimized HALC$^\dagger$ generates short answers, it took 7 times more time than the original due to the complexity of the method itself. For CRG, we include the time cost for object detection while we use provided external data. VSCoDe does not use an external model or dataset, and especially using *coreset*, the time and memory cost is similar to that of single CD methods.

While HALC and CRG show high performance on some tasks, the time and memory cost of using an external model is expensive. In addition, the performance of the two methods can be affected depending on which object detector is used. In the case of CRG, the MSCOCO dataset was used in the pre-training process of GroundingDINO-B, the object detector, resulting in high performance on POPE of MSCOCO dataset.

**Results on various tasks.** As shown in Figure 5, VSCoDe demonstrates superior performance across both VQA and captioning tasks. The figure shows that VSCoDe consistently outperforms other CD methods across diverse VQA task categories and answer formats, including benchmarks such as MME, MMBench, VQAv2, and POPE. Furthermore, it generates significantly fewer hallucinated answers, as evidenced by multiple performance metrics on the captioning task and the POPE benchmark. In particular, higher BLEU scores indicate that the generated captions more accurately describe the given image, while lower CHAIR scores demonstrate a reduction in hallucinated content.

Figure 5: Performance of the LLaVA-7B model on various VQA and captioning tasks. In most tasks, VSCoDe shows the best performance.

### 4.3 Discussion

**Analysis of the validity of CD selection by distance.** Table 1 shows that choosing appropriate visual augmentation for each question is essential to maximize contrastiveness for an effective CD. In this section, we analyze the effectiveness of using distance $D(\cdot)$ for augmentation selection. Figure 6 shows the proportion of

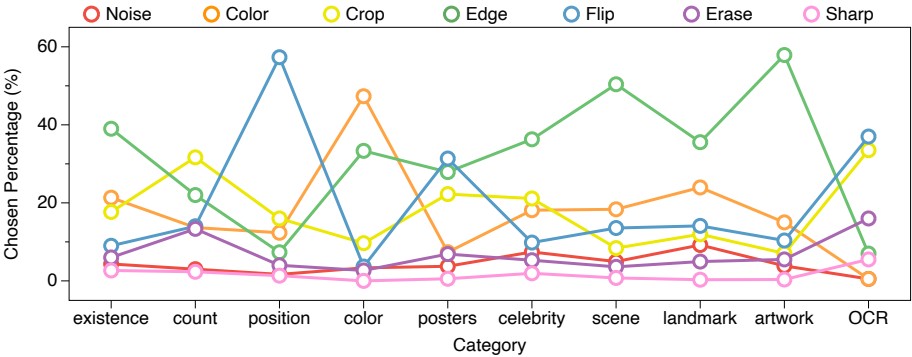

Figure 6: The percentage of each augmentation being selected based on the distance $D(\cdot)$ within each category on MME benchmark.

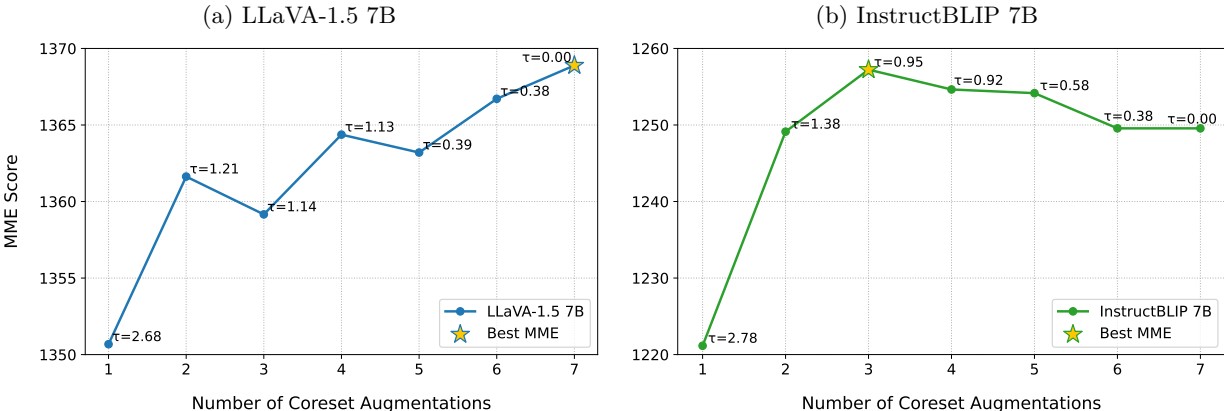

Figure 7: Ablation study on the effect of acceptance threshold $\tau$ in the *coreset* strategy across LLaVA-1.5 7B and InstructBLIP 7B. Each point on the curves indicates the $\tau$ threshold that determines the number of augmentations selected into `VSCoDe`, corresponding to the value shown on the $x$-axis. The $\tau$ yielding the highest performance is denoted as *Best MME*.

single augmentations selected by `VSCoDe` for each category of MME when distance $D(\cdot)$ is used. Accordingly, $D(\cdot)$ can effectively select visual augmentations that significantly increase contrast for each category. For example, the 'color' category has a high proportion of color augmentation, which distorts the image's color, and the 'position' category has a considerable proportion of flip augmentation, which distorts the position information of the given image. Furthermore, $D(\cdot)$ goes beyond simply using a single augmentation for queries of a specific category. According to Table 2, for many categories, `VSCoDe` outperforms each augmentation. This means that distance improves effective augmentation selecting for CD on a single category, beyond simply using a single high-contrastiveness augmentation. More detailed analysis can be found at Appendix D.

**Effect of *coreset* strategy.** As shown in Table 2 and Figure 5, although *coreset* uses less number of augmentations than *all*, it performs better in many cases. This indicates that our *coreset* strategy, which removes noisy augmentations, is effective and underscores the importance of selecting only the most impactful augmentations for better results. This method not only enhances both efficiency and effectiveness but also serves as a useful tool for choosing augmentations from multiple options. Table 4 presents the selection ratio of augmentations by `VSCoDe` across different datasets. According to the strategy, when 7 augmentations are used in the set, each must have a selection ratio of at least 7.14% to be included. This demonstrates that the most impactful augmentation types vary depending on the data distribution. For instance, in the captioning task, which requires accurate image description, augmentations like the crop—which removes the background regions of an image—have a higher impact on it, while the effects of noise and color are relatively diminished.

Table 4: Selection percentage (%) of `VSCoDe` across datasets and augmentation types. Candidates excluded by the coreset strategy threshold $\tau$ are highlighted in gray.

| Dataset | Augmentation Types | | | | | | |
|---|---|---|---|---|---|---|---|
| | Noise | Color | Crop | Edge | Flip | Erase | Sharp |
| MME | 5.42 | 17.00 | 16.00 | 40.16 | 15.72 | 4.96 | 0.74 |
| MMBench | 5.04 | 13.84 | 16.45 | 36.27 | 21.51 | 5.82 | 1.09 |
| VQAv2 | 7.38 | 18.77 | 14.51 | 40.82 | 12.16 | 5.22 | 1.14 |
| POPE(MSCOCO) | 8.03 | 15.03 | 17.40 | 38.83 | 13.03 | 5.43 | 1.74 |
| POPE(AOKVQA) | 5.90 | 15.43 | 16.37 | 40.20 | 14.67 | 5.63 | 1.80 |
| POPE(GQA) | 3.00 | 15.00 | 18.27 | 42.23 | 13.77 | 5.93 | 1.80 |
| Captioning | 1.60 | 6.80 | 28.20 | 43.00 | 14.00 | 6.00 | 0.40 |

Table 5: Comparison of augmentation selection on the first token and full tokens at the captioning task. Dist (F) denotes selecting CD augmentation based on the first token, whereas Dist (E) re-selects CD augmentation at every token.

| Method | Augmentation | | | | Score | | | Cost | |
|---|---|---|---|---|---|---|---|---|---|
| | Type | Q-Dep | Select | Metric | BLEU | CHAIR$_S$ | CHAIR$_I$ | Time | Mem |
| Vanilla | - | - | - | - | $40.13_{\pm1.10}$ | $8.00_{\pm0.82}$ | $5.47_{\pm0.59}$ | 1.00 | 1.00 |
| Random | All | ✗ | ✓ | Rand | $49.02_{\pm1.38}$ | $5.75_{\pm0.83}$ | $3.91_{\pm0.66}$ | 1.96 | 1.00 |
| VSCoDe | *All* | ✓ | ✓ | Dist (F) | $51.37_{\pm0.56}$ | $4.80_{\pm0.77}$ | $3.21_{\pm0.59}$ | ×2.45 | ×1.04 |
| | *Coreset* | ✓ | ✓ | Dist (F) | $\mathbf{51.97}_{\pm0.48}$ | $4.60_{\pm0.56}$ | $3.10_{\pm0.40}$ | ×2.15 | ×1.03 |
| VSCoDe-E | *All* | ✓ | ✓ | Dist (E) | $\underline{51.75}_{\pm0.45}$ | $\mathbf{4.20}_{\pm0.33}$ | $\mathbf{2.83}_{\pm0.13}$ | ×3.34 | ×1.41 |
| | *Coreset* | ✓ | ✓ | Dist (E) | $51.61_{\pm1.17}$ | $\underline{4.24}_{\pm0.87}$ | $\mathbf{2.83}_{\pm0.55}$ | ×2.69 | ×1.37 |

Table 6: MME performance of `VSCoDe` with different candidate combinations. We evaluate the performance of the candidate set $\mathcal{A}$ by excluding each candidate one by one.

| Method | Augmentation | | | | Category | | | | | | | | | | |
|---|---|---|---|---|---|---|---|---|---|---|---|---|---|---|---|
| | Type | Q-Dep | Select | Metric | Existence | Count | Position | Color | Posters | Celebrity | Scene | Landmark | Artwork | OCR | Total |
| Vanilla | - | - | - | - | 180.00 | 112.00 | 117.67 | 147.00 | 121.02 | 110.24 | 148.80 | 129.15 | 109.35 | 97.00 | 1327.55 |
| Single CD | Color | ✗ | ✗ | - | 182.00 | 134.00 | 129.33 | 160.00 | 142.86 | 142.24 | 154.60 | 143.40 | 112.60 | 113.50 | 1414.53 |
| | Crop | ✗ | ✗ | - | 187.00 | 110.33 | 138.33 | 147.67 | 149.80 | 146.65 | 156.70 | 146.65 | 105.75 | 103.50 | 1392.38 |
| | Flip | ✗ | ✗ | - | 183.00 | 122.00 | 129.00 | 155.00 | 143.61 | 132.12 | 151.45 | 133.90 | 109.55 | 115.00 | 1374.62 |
| VSCoDe | $\mathcal{A}_{\{color,crop\}}$ | ✓ | ✓ | Dist | 186.00 | 116.67 | 132.33 | 160.00 | 150.27 | 149.82 | 155.70 | 153.35 | 108.75 | 108.00 | $\underline{1420.90}$ |
| | $\mathcal{A}_{\{color,flip\}}$ | ✓ | ✓ | Dist | 181.00 | 138.33 | 136.33 | 161.67 | 145.10 | 141.41 | 150.10 | 141.00 | 113.55 | 108.00 | 1416.50 |
| | $\mathcal{A}_{\{crop,flip\}}$ | ✓ | ✓ | Dist | 184.00 | 116.00 | 133.33 | 150.67 | 148.57 | 147.94 | 155.55 | 151.50 | 107.80 | 103.50 | 1398.86 |
| | $\mathcal{A}_{\{color,crop,flip\}}$ | ✓ | ✓ | Dist | 183.00 | 120.33 | 133.33 | 161.00 | 150.07 | 149.94 | 155.70 | 155.70 | 109.20 | 108.00 | $\mathbf{1426.28}$ |

**Effect of acceptance threshold $\tau$ on the *coreset* strategy.** We conduct an ablation study on the acceptance threshold $\tau$ to investigate its role in the Coreset Strategy. As described in Section 3.4, a lower $\tau$ allows a greater number of augmentations to participate in `VSCoDe`, which increases computational cost while improving performance. In contrast, a higher $\tau$ filter out less effective augmentations and retains only those with more substantial contributions, thus reducing memory and time requirements. In particular, augmentations excluded at higher $\tau$ values tend to provide larger gains when included in `VSCoDe`. In contrast, those admitted only at lower thresholds contribute marginal improvements or can even degrade performance. As shown in Figure 7, this trade-off is consistently observed in both LLaVA-1.5 7B and InstructBLIP 7B. Interestingly, in the case of InstructBLIP, using the Coreset with an appropriate $\tau$ can even outperform the total augmentation set, highlighting that efficiency and accuracy gains can be achieved simultaneously. In general, these results demonstrate that carefully tuned $\tau$ within the Coreset Strategy not only improves computational efficiency but also provides robustness in performance in `VSCoDe`.

**Selecting augmentation $\hat{o}$ on the first token.** `VSCoDe` mainly uses the first token to select the augmentation. Selecting the augmentation on every token improves performance, but it comes from an additional cost. In this case, the memory cost is severe but can reduce time by running in a batch-wise manner. Without batch-wise operation, the time cost increases linearly depending on the number of augmentation candidates. We name this strategy that CD on every token as `VSCoDe-E` and report the result in Table 5 on captioning task using the MSCOCO benchmark. While it needs about 130% more time and memory, it shows about 115% more performance gain in CHAIR metrics compared to `VSCoDe`. Using contrastive augmentation on each token may be more helpful to mitigate hallucination.

**Analysis of the combination of visual augmentations.** We evaluate different combinations of augmentations to estimate the impact of each augmentation. For simplicity, we limit the augmentation set to {`color`, `flip`, `random crop`} for this section. Table 6 shows the effect of using all augmentation candidates in the set and the impact of excluding each one individually. According to the results, `VSCoDe` performance using all three augmentations, color, crop and flip, shows higher performance than other sub-combinations. Specifically, when color or flip augmentation is removed from the augmentation set, performance in the color or position categories significantly decreases. Considering each augmentation has a different contrastive effect,

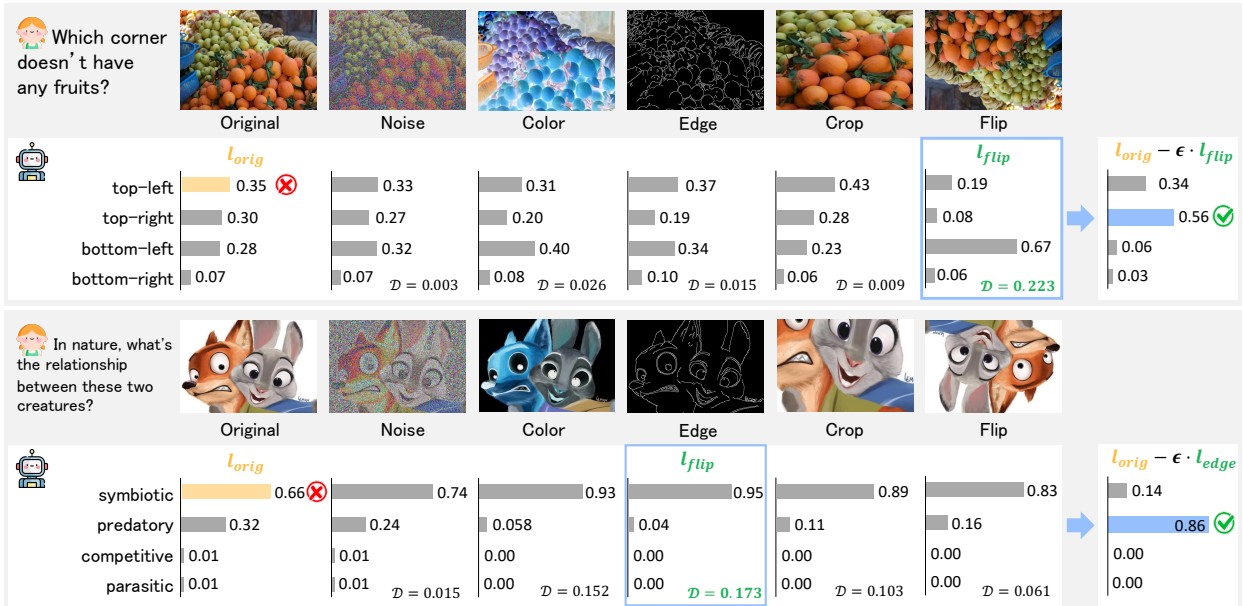

Figure 8: Two qualitative examples of `VSCoDe` on MMBench benchmark. On both cases, `VSCoDe` successfully corrects the wrong prediction from the original image by selecting the augmentation with biggest distance $D(\cdot)$.

the results confirm that providing an appropriate combination of VAs for $\mathcal{A}$ to cover all types of queries can provide proper contrast for a given task.

**Qualitative study.** In this section, we discuss examples of applying `VSCoDe` on MMBench with LLAVA-13B as illustrated in Figure 8. The first example shows that the original image yields an incorrect prediction, identifying an empty space as 'top-left'. To adjust the logit of the ground truth token through CD, the augmented image must have a low logit for the ground truth token. Only the flipped image introduces positional contrast, resulting in a low logit for the token and enabling a successful CD. On the second example, the incorrect answer already holds a high probability, indicating that strong contrastiveness is needed to correct the prediction. Although all augmentations increase the logit of the wrong answer token, adding noise to the image produces only a minor difference, which keeps the prediction incorrect. In this situation, maximizing contrastiveness is essential to achieve the correct answer. Therefore, selecting the augmentation with bigger $D(\cdot)$ is effective in reaching the correct answer. The CD results from unselected augmentations can be found in Appendix F.

# 5 Related Works

**Contrastive decoding.** CD (Li et al., 2023b) was introduced in the NLP domain using two differently sized language models. It leverages contrastive output by subtracting the small model's probability from the larger model's to retain the strengths of the large model while eliminating the weaknesses that are evident in the small model. There are variants like DOLA (Chuang et al., 2023) which utilizes contrast in layer-level outputs and Instructive Decoding (Kim et al.) uses two contrastive instructions to generate an output opposite to the original output.

Recently, similar approaches have been applied in LVLMs, utilizing contrastive images to guide the model in generating accurate text, mainly focusing on reducing hallucination in LVLM (Li et al., 2023c; Liu et al., 2023a; Tong et al., 2024). VCD (Leng et al., 2023) demonstrates that adding noise to the image can elevate the hallucination inherent in LVLMs, subsequently applying CD to manage the hallucination. Another work CRG (Wan et al., 2024) employs a black bounding box from external data to conceal the object relevant to the question, amplifying hallucination, while HALC (Chen et al., 2024) uses multiple different cropped images from the detection model and explores multiple pairs of cropped images to find pairs that amplify the information in the cropped image. These works address methods to manage hallucination in LVLMs

using a single type of augmentation, which has limitations in generating enough contrast for various types of questions. Unlike previous studies, `VSCoDe` explores multiple augmentations and selects the most effective one to answer the question. Moreover, it does not require additional training or an external model, providing direct perturbation to the image.

**Visual augmentation.** In the computer vision domain, visual augmentation has been employed to increase the diversity of sample data, thereby helping to overcome the challenges associated with acquiring large training datasets and mitigating overfitting issues in environments with limited samples. Traditional augmentations include changes in color, cropping, and flipping. Additionally, there are more advanced techniques such as erasing (Kumar Singh & Jae Lee, 2017; DeVries & Taylor, 2017; Zhong et al., 2020), and other techniques such as mixup (Zhang et al., 2017) and CutMix (Yun et al., 2019). Furthermore, the automatic application of multiple augmentations has been explored (Cubuk et al., 2019; Lim et al., 2019).

Some studies in LVLMs employ VA to achieve the desired output in various methods. FGVP (Yang et al., 2024) adds blur to the background of the image, leaving the main object clear to emphasize it. To focus on each object in the image, (Chen et al., 2023; Surís et al., 2023; Lin et al., 2024) use multiple cropped images, each focusing on a single object to generate the desired output, while (Kim et al., 2023) uses inpainting to erase objects to measure the correlation between objects.

## 6 Conclusion

In this paper, we present `VSCoDe`, a method that leverages multiple augmentations by adaptively selecting the most contrastive augmentation for effective CD. Our initial analysis revealed that the level of contrast generated by augmentations varies according to the query, so choosing an appropriate augmentation is critical for enhancing CD. Building on this insight, we developed `VSCoDe`, an algorithm that selects augmentations based on the largest distance $D(\cdot)$ metric. Experimental results demonstrate that `VSCoDe` surpasses existing methods across various tasks, highlighting the importance of using appropriate augmentations.

**Limitation.** Our method selects the appropriate contrastive augmentation among augmentation candidates. No matter how well `VSCoDe` works and the appropriate augmentation is chosen for the given task, if there is no sufficient contrastive augmentation for the task among the candidates, it is difficult to expect a significant performance gain.

**Future Work.** For future work, exploring methods to determine task-specific augmentation candidates could be a valuable direction. Additionally, identifying a subset of highly influential augmentations or leveraging augmentation sets proposed in other works could be a promising direction. Furthermore, investigating the impact of text augmentations in conjunction with visual augmentations and integrating the two approaches could further improve the results.

## Acknowledgement

This research was supported by the Institute of Information & communications Technology Planning & Evaluation (IITP) grant funded by the Korea government (MSIT) (No. 2022-0-00871, Development of AI Autonomy and Knowledge Enhancement for AI Agent Collaboration, 50%) and the Ministry of Trade Industry & Energy (MOTIE, Korea), under the project titled "An Unbreachable Multilayer AI-based Security Mechanism that Continuously Adapts and Evolves in Dynamic Conditions" (Project Number: RS-2025-02653102, 45%) and Institute for Information & communications Technology Planning & Evaluation(IITP) grant funded by the Korea government(MSIT) (RS-2019-II190075, Artificial Intelligence Graduate School Program(KAIST), 5%).

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

-Supplementary Material-

# VSCoDe: Visual-Augmentation Selection for Contrastive Decoding

## A  Ablation on Distance metric $D(\cdot)$

In this section, we examine comprehensive several additional ablation experiments that are considerable in the environment in which `VSCoDe` is applied. Based on these ablation results, we expect `VSCoDe` to have universally high robustness and be able to perform various tasks, models, and inferences.

We perform experiments using several common distance measures to define our function $D(\cdot)$ that `VSCoDe` uses to select which VA will produce high contrast. The experiment is performed in the MME dataset using the `LLaVA-1.5` 13B model. Also, we use the average softmax `Gain` directly to check the effect. In detail, `Gain` on the correct answer label obtained when applying the distance measure candidate $D_i$ used in the experiment and the VAs used in Figure 2 to `VSCoDe` for all samples. In order to control the variables of VAs that contain randomness, each experiment performs a total of 5 experiments with different seeds on the entire `MME` dataset and then measures `Gain` through the average.

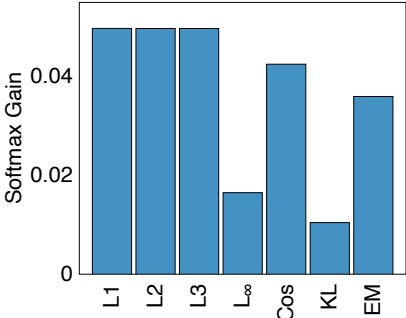

Figure 9: Average softmax gain by different distance metrics.

Figure 9 shows the result of affectness of different $D(\cdot)$ metrics. In this experiment, we use $L_1$, $L_2$, $L_3$, $L_\infty$, Cosine similarity, Kullback-Leibler Divergence (KL divergence), and Earth Mover's distance (EM distance) as candidates. The x-axis of the results in Figure 9 means the candidate distance names used, and the y-axis means the average softmax gain improved compared to regular decoding obtained through `VSCoDe` when each distance is used as a measurement. From the results, we can check that $L_1$, $L_2$, and $L_3$ norms show high performance improvement almost no difference overall. This means that any of these can be used in the algorithm as a distance function at a similar level. However, in the case of $L_\infty$ and KL divergence, it can be seen that the actual performance improvement is much smaller compared to others. These show very low-performance improvement compared to the $L_2$ distance, which we used in the main experiment, meaning they are improper measurements for estimating the expected contrast of VAs. The other two distances, cosine similarity, and EM distance, performed higher than KL divergence but did not perform higher than $L_2$ norm for the entire `MME` dataset. Based on this result, we empirically confirmed that using $L_2$ norm as our main $D(\cdot)$ for `VSCoDe` is a meaningful standard through experiments with these distance measures and the results shown throughout our main experiments.

## B  Analysis of Different Model Types and Sizes

We showed that `VSCoDe` is proper for general LVLMs and has a significant effect on performance by experimenting with five different models `LLaVA-1.5`, LLaVA-OneVision, InstructBLIP, Qwen-VL, and Qwen2-VL on various types of datasets at the Section 4. In this ablation, we conduct an experiment using `LLaVA-1.5` 7B, 13B and InstructBLIP 7B, 13B to check the effect of the model size on `VSCoDe`. MME dataset is used for

Table 7: MME performance by different models.

| Method | Augmentation | | | | Model | | | | |
| | Type | Q-Dep | Select | Metric | LLaVA-1.5 7B | LLaVA-OneVision 7B | InstructBLIP 7B | Qwen-VL 7B | Qwen2-VL 7B |
|---|---|---|---|---|---|---|---|---|---|
| Vanilla | - | - | - | - | 1272.22 | 1441.59 | 1155.26 | 1355.32 | 1481.82 |
| Single CD | Color | ✗ | ✗ | - | 1347.24 | 1528.28 | 1224.26 | 1422.69 | 1513.15 |
| | Edge | ✗ | ✗ | - | 1350.68 | 1547.04 | 1221.15 | 1393.32 | 1552.14 |
| | Sharp | ✗ | ✗ | - | 1323.60 | 1495.46 | 1177.84 | 1395.14 | 1516.84 |
| | Crop | ✗ | ✗ | - | 1338.50 | 1517.61 | 1194.13 | 1396.83 | 1512.10 |
| | Erase | ✗ | ✗ | - | 1310.89 | 1509.56 | 1195.27 | 1385.33 | 1501.28 |
| | Flip | ✗ | ✗ | - | 1344.75 | 1530.90 | 1222.34 | 1425.20 | 1541.49 |
| VCD | Noise | ✗ | ✗ | - | 1323.44 | 1531.79 | 1218.90 | 1406.15 | 1519.61 |
| VSCoDe | *All* | ✓ | ✓ | Dist | **1368.89** | **1579.34** | 1249.56 | 1406.05 | 1547.07 |
| | *Coreset* | ✓ | ✓ | Dist | 1364.36 | 1574.25 | **1254.16** | **1426.43** | **1552.91** |

Table 8: MME performance by different model sizes.

| Method | Augmentation | | | | Model | |
| | Type | Q-Dep | Select | Metric | LLaVA-1.5 13B | InstructBLIP 13B |
|---|---|---|---|---|---|---|
| Vanilla | - | - | - | - | 1327.55 | 1151.45 |
| Single CD | Color | ✗ | ✗ | - | 1414.53 | 1237.71 |
| | Edge | ✗ | ✗ | - | 1424.20 | 1220.63 |
| | Sharp | ✗ | ✗ | - | 1361.74 | 1164.32 |
| | Crop | ✗ | ✗ | - | 1384.65 | 1205.55 |
| | Erase | ✗ | ✗ | - | 1365.91 | 1185.32 |
| | Flip | ✗ | ✗ | - | 1374.62 | 1213.61 |
| VCD | Noise | ✗ | ✗ | - | 1354.34 | 1208.44 |
| VSCoDe | *All* | ✓ | ✓ | Dist | 1443.06 | 1248.30 |
| | *Coreset* | ✓ | ✓ | Dist | **1443.14** | **1256.09** |

this experiment. We measured the performance for the perception category and the total performance for each model.

Table 7 and Table 8 show the performance of `VSCoDe` on each model and size for the `MME` dataset. From the result, we can confirm even if the model size and model used are different, the softmax gain obtained when each VA is used in `VSCoDe` is robust to the type and size of the model and shows a tendency to be dependent on the given task. Throughout the experimental results, the single VA `edge` and `color` show very high performance. On the other hand, we can see that single VA `sharp` and `erase` have a slight performance gain than others. For different models, the performance gain shown by each VA shows an overall similar trend, and there is a higher performance improvement compared to the original regular decoding.

Furthermore, for different model sizes, there is a significant performance gain when applying our algorithm `VSCoDe`. `VSCoDe` using all of the VAs specified in Figure 2 shows a higher performance improvement than using each single VA. This indicates that, regardless of model and size, each application has the highest performance in the entire perception category and total performance.

## C Analysis of Different Sampling Strategies

We perform analysis studies on different sampling strategies to see how `VSCoDe` is affected by sampling methods other than basic regular decoding. In this experiment, 4 sampling techniques are applied: (1) Top P sampling (specifically, $p = 0.9$), (2) Top K sampling (specifically, $k = 50$), (3) Temperature sampling (specifically, $T = 0.7/1.5$). Top P sampling is a method in which the only token candidates in the distribution on cumulative probability $p$ can be selected as the next token. This has the effect of preventing noise samples with too low a probability to be extracted from candidates. Top K sampling uses only the top $k$ candidates from the

Table 9: MME performance by different sampling strategies for LLaVA-1.5 13B.

| Method | Augmentation | | | | Sampling Strategy | | | |
|---|---|---|---|---|---|---|---|---|
| | Type | Q-Dep | Select | Metric | Top P ($p$=0.9) | Top K ($k$=50, $T$=0.7) | Temp ($T$=0.7) | Temp ($T$=1.5) |
| Vanilla | - | - | - | - | 1352.87 | 1399.33 | 1403.99 | 1169.71 |
| Single CD | Color | ✗ | ✗ | - | 1405.90 | 1443.27 | 1445.19 | 1349.20 |
| | Edge | ✗ | ✗ | - | 1434.14 | 1433.86 | 1420.64 | 1364.72 |
| | Sharp | ✗ | ✗ | - | 1381.00 | 1415.88 | 1416.63 | 1294.08 |
| | Crop | ✗ | ✗ | - | 1391.01 | 1413.09 | 1422.15 | 1342.43 |
| | Erase | ✗ | ✗ | - | 1374.47 | 1404.27 | 1399.67 | 1315.08 |
| | Flip | ✗ | ✗ | - | 1404.76 | 1426.97 | 1425.54 | 1340.88 |
| VCD | Noise | ✗ | ✗ | - | 1370.47 | 1425.60 | 1429.52 | 1316.95 |
| VSCoDe | *All* | ✓ | ✓ | Dist | **1462.67** | 1456.03 | 1454.32 | **1389.03** |
| | *Coreset* | ✓ | ✓ | Dist | 1462.58 | **1457.10** | **1458.73** | 1377.47 |

highest probability for sampling. In temperature sampling, temperature scaling is applied to the softmax to calculate the next token logits. When temperature $T$ is low, the possibility of selecting a high-probability candidate group increases, and the possibility of choosing low-probability candidates decreases. It has the effect of increasing the probability of more static responses. Conversely, when the temperature $T$ is large, the chance of choosing among the high-probability candidates decreases, and the low-probability candidates increases. It has the effect of increasing the possibility of making more diverse responses.

Table 9 show the experiment result of VSCoDe with different sampling strategies. From the table, we can check that VSCoDe gives us a high performance in various types of samplings. This is not only for regular decoding, but it also shows higher performance compared to single VA in the Top P sampling and Top K sampling. A notable observation is that VSCoDe shows high performance in both cases where the temperature scale gets higher or lower. In the case of high temperature, the model has a higher probability of generation more diverse, and the explanations and representations are getting richer. However, in this case, there is a potential problem that the entire output is inaccurate while in generation. In particular, if specific information for a given image must be utilized rather than using inherent prior knowledge, there is a possibility that incorrect output may lose correlation with visual information on LVLMs. Our results show that using VSCoDe in this situation can be expected to have the effect of concentrating the model to intentionally utilize visual information by contrastive decoding the output through contrast VA. As can be seen from the results, in situations where the temperature scale is large, CD through VA produces a more significant performance gain. Additionally, the magnitude of contrastiveness produced by each VA is different in the task so we can see a considerable performance difference between single VA CDs. In this situation, VSCoDe, which automatically selects and applies the appropriate VA for a given task, can be used more appropriately and robustly to the given scenario. Furthermore, it shows that VSCoDe has the highest performance improvement.

Our algorithm can also be used in low-temperature scale scenarios, which increases the sampling possibility of a high-probability token being chosen as the next token. In this scenario, the original model's high logits become more extensive than usual by temperature scaling, increasing the probability of being selected as the next token. When the correct answer logit does not have a high value, the possibility of being chosen as the next token is crucially dropped. For a low-temperature scale, once the model starts generation with an incorrect token, it is more likely to continue generating incorrect responses. As mentioned in CD, in the case of high confidence in high logit sampling methods in a generation, a wrong token selection can significantly impact the quality of future responses. In this situation, using VSCoDe can increase the likelihood that a low correct answer token will be selected as the correct answer through CD using contrast VA. As a result, it shows high robustness against the temperature sampling scale and increases the likelihood of providing an appropriate response.

Table 10: MME score on perception tasks using the LLaVA-1.5 13B. The best and second-best performances are reported using **bold** and underline formatting, respectively. As with LLaVA-1.5 7B, `VSCoDe` shows strong performance on the 13B model.

| Method | Augmentation | | | | | Category | | | | | | | | | | Total |
|---|---|---|---|---|---|---|---|---|---|---|---|---|---|---|---|---|
| | Type | Q-Dep | Ext | Select | Metric | Existence | Count | Position | Color | Posters | Celebrity | Scene | Landmark | Artwork | OCR | |
| Vanilla | - | - | - | - | - | 182.00 | 125.33 | 110.33 | 154.67 | 128.57 | 123.00 | 153.05 | 131.30 | 108.30 | 111.00 | $1327.55_{\pm16.2}$ |
| Single CD | Color | ✗ | ✗ | ✗ | - | 182.00 | 134.00 | 129.33 | 160.00 | 142.86 | 142.24 | 154.60 | 143.40 | 112.60 | 113.50 | $1414.53_{\pm9.56}$ |
| | Edge | ✗ | ✗ | ✗ | - | 185.00 | 146.00 | 125.00 | 157.67 | 141.70 | 142.24 | 152.95 | 139.50 | 113.15 | 121.00 | $1424.20_{\pm22.0}$ |
| | Sharp | ✗ | ✗ | ✗ | - | 182.00 | 113.33 | 130.00 | 156.33 | 136.46 | 130.76 | 156.90 | 137.10 | 109.85 | 109.00 | $1361.74_{\pm20.3}$ |
| | Crop | ✗ | ✗ | ✗ | - | 183.00 | 127.00 | 124.33 | 150.00 | 143.13 | 139.24 | 153.05 | 141.70 | 108.70 | 114.50 | $1384.65_{\pm24.7}$ |
| | Erase | ✗ | ✗ | ✗ | - | 185.00 | 126.67 | 116.33 | 144.67 | 147.55 | 128.29 | 156.60 | 132.85 | 110.95 | 117.00 | $1365.91_{\pm22.5}$ |
| | Flip | ✗ | ✗ | ✗ | - | 183.00 | 122.00 | 129.00 | 155.00 | 143.61 | 132.12 | 151.45 | 133.90 | 109.55 | 115.00 | $1374.62_{\pm14.9}$ |
| VCD | Noise | ✗ | ✗ | ✗ | - | 185.00 | 122.33 | 125.00 | 151.67 | 137.62 | 133.12 | 151.15 | 139.10 | 110.85 | 98.50 | $1354.34_{\pm24.5}$ |
| ICD | Text | ✗ | ✗ | ✗ | - | 182.00 | 124.00 | 127.33 | 159.00 | 134.63 | 126.47 | 156.00 | 137.00 | 110.45 | 103.00 | $1359.88_{\pm16.78}$ |
| CRG | Erase | ✓ | ✓ | ✗ | - | 187.00 | 109.67 | 129.33 | 147.67 | 147.35 | 140.76 | 162.15 | 153.35 | 105.15 | 116.00 | $1398.43_{\pm12.46}$ |
| Random | *All* | ✗ | ✗ | ✓ | Rand | 183.00 | 118.33 | 135.00 | 153.67 | 141.29 | 133.06 | 151.45 | 140.95 | 110.40 | 106.00 | $1373.15_{\pm28.09}$ |
| `VSCoDe` | *All* | ✓ | ✗ | ✓ | Dist | 184.00 | 138.67 | 134.00 | 167.00 | 146.80 | 144.29 | 149.35 | 145.30 | 114.65 | 119.00 | $1443.06_{\pm6.80}$ |
| | *Coreset* | ✓ | ✗ | ✓ | Dist | 183.00 | 140.33 | 132.00 | 165.33 | 146.46 | 143.71 | 149.80 | 145.05 | 114.45 | 123.00 | $\mathbf{1443.14_{\pm9.99}}$ |

# D Analysis of the Selected Augmentations

In this section, we discuss how often each augmentation is selected by `VSCoDe` based on the maximum distance D within each category of the MME benchmark using LLaVA-1.5 13B. The corresponding MME benchmark performance of using each augmentation and `VSCoDe` for CD is reported in Table 10, and the selection ratio of each augmentation by categories can be seen at Figure 6. In most categories, `edge` is predominantly selected, which aligns with its high performance, as shown in the table. Moreover, `flip` and `color` are mostly selected in the position and color categories, respectively, which is intuitive since `flip` and `color` augmentations alter position- and color-related information. `Noise` and `sharp` augmentations, which show low performance in the table, are selected less frequently, as there are other more effective augmentations available.

# E Experiment Details

## E.1 Experimental Setting

Here, we provide detailed description about the setting and results of the experiments with an additional baseline and computational cost. We compare with VCD (Leng et al., 2023) ICD (Wang et al., 2024), HALC (Chen et al., 2024), and CRG (Wan et al., 2024) and follow the setting of original paper for the experiments. VCD (Leng et al., 2023) is a method that applies contrastive decoding to an image manipulated by adding diffusion noise. It adds Gaussian diffusion noise to a given image with a total of 500 steps. ICD uses a contrastive query that adds 'You are a confused object detector' to the head of the given query text.

HALC (Chen et al., 2024) is a method that uses an additional object detector to contrastively decode an adaptive sample for a given query image called Adaptive Focal-Contrast Grounding. We used the official HALC code to reproduce results with the default setting using GroundingDINO as object detector (Liu et al., 2023c). It provides the codes for MME, MSCOCO captioning, and POPE COCO, which we used to reproduce the results. However, in this process, although the reproduced performance of HALC was similar to the values reported in the HALC paper, the scores were significantly lower compared to our baselines. In addition, HALC uses a unique evaluation metric called OPOPE, which was unavailable for conducting an accurate comparison.

Therefore, we conducted various types of prompt engineering to bring out the potential of the HALC method and achieve a similar level. However, HALC did not perform well in prompt instruction following. Using the default prompt texts we applied to other baselines or common prompts used in VLM resulted in poor performance and generated output sentences with unwanted lengthy interpretations. Consequently, we obtained captioning output using a prompt 'provide a short description for this photo without interpretations.',

which was almost the only prompt that could force the output to answer with a short caption. The result of the performance obtained was recorded as HALC$^\dagger$ at Table 3.

We provide CRG (Wan et al., 2024) for an additional baseline, which is a method that uses an image with the corresponding region of interest (RoI) removed from the image given in the query for contrastive decoding. If no specific RoI is given, CRG extracts noun phrases from the given query text and gets the RoI of extracted phrases by using the object detector. Following the original paper, we use spaCy (Honnibal et al., 2020) for the noun phrase detector and obtained bounding boxes by using GroundingDINO-B (Liu et al., 2023c) with a threshold of 0.3. In this process, we filtered noun phrases that do not represent objects ("the image", "the type", "the photo", etc.) in the prompt of MME or POPE benchmark to prevent incorrect RoI extraction. For the MSCOCO captioning task, we use bounding boxes provided by the annotation of the dataset.

In this paper, all reports of our experiment used LVLM models that can run on a single 48 GB NVIDIA RTX A6000.

## F  More Case Studies

We present additional examples of MMBench using the LLaVA-1.5 13B model. The figures illustrate the probability distribution of outputs from original and augmented images, along with the probabilities after applying CD. The yellow bar represents the option with the highest probability for the original image, the green bar shows the top option for the augmented image with the maximum distance $D(\cdot)$, and the blue bar indicates the option with the highest probability which is correctly aligned with the ground truth after applying CD.

Figure 10c and Figure 10d demonstrate that augmentations with maximum $D(\cdot)$ effectively manipulate the image to produce incorrect predictions. Note that due to hallucination, the model retains a non-negligible probability for the original prediction. CD mitigates this hallucination, enabling the model to generate correct outputs. Conversely, as shown in Figure 10e, Figure 10f, and Figure 10b, augmentations with maximum D amplify the incorrect predictions of the original image, facilitating successful CD. Figure 10g highlights that, although the augmented images yield high confidence for the same option as the original image, which corresponds to the ground truth option, CD leverages only reliable tokens with subsequent probabilities $V_{\text{cand}}$, as explained in Section 3.3, to produce trustworthy answers. In this case, only option D is considered as $V_{\text{cand}}$ since other options have low logit values. Finally, Figure 10a illustrates a failure case where the output distribution of the original image is uniformly spread across options, causing ambiguity in determining the direction of image manipulation that facilitates the incorrect output. Although the probability of a correctly predicted output turning into an incorrect one is only 3.5%, this challenge remains in future research.

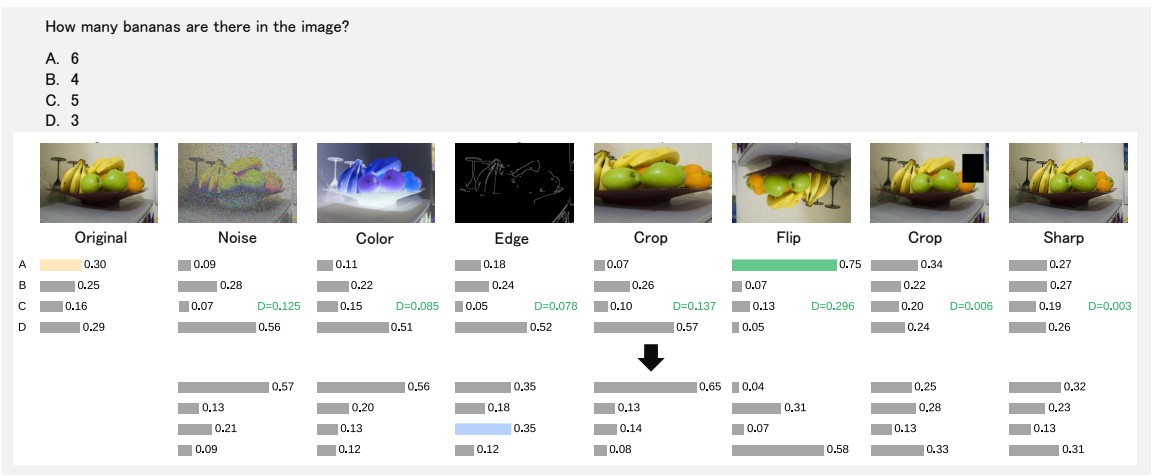

(a) Question: "How many bananas are there in the image?"

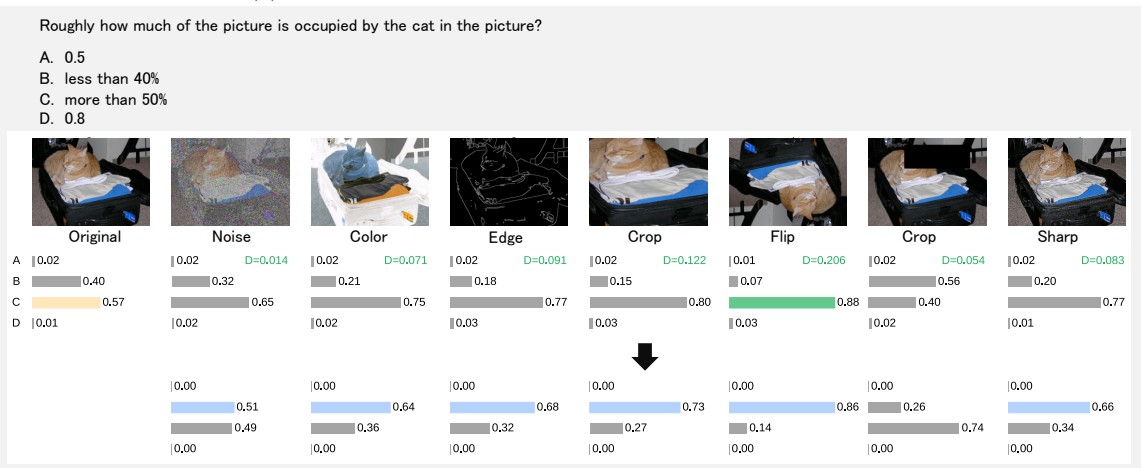

(b) Question: "Roughly how much of the picture is occupied by the cat?"

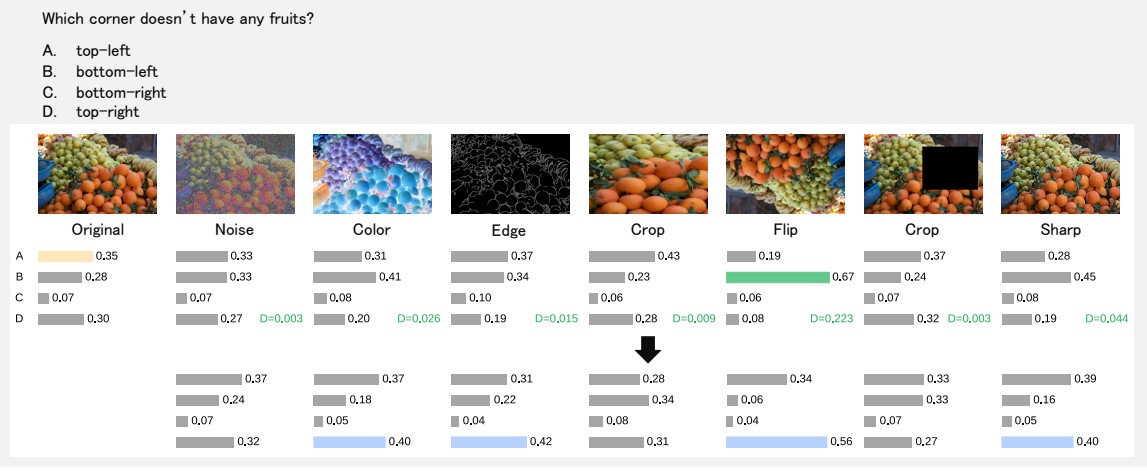

(c) Question: "Which corner doesn't have any fruits?"

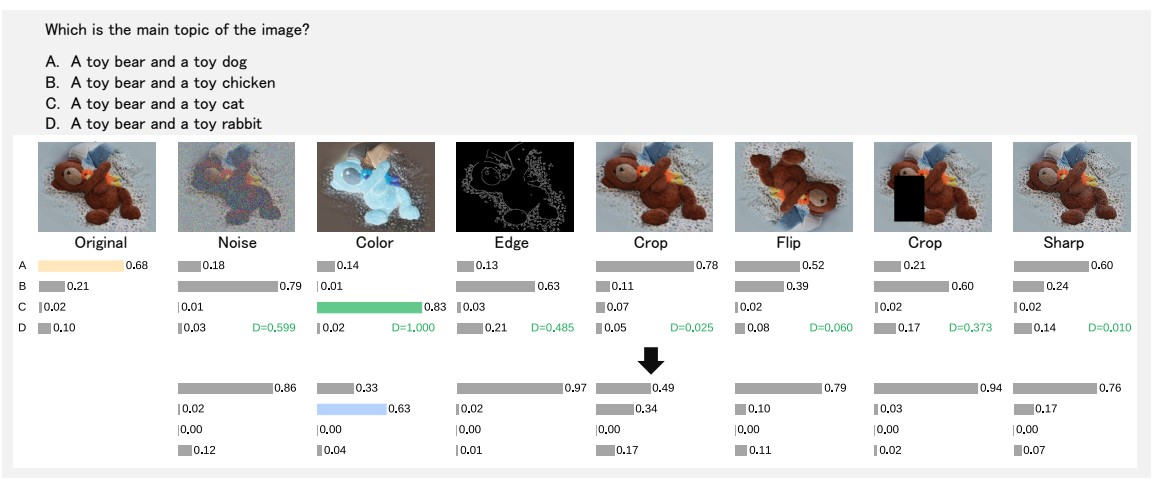

(d) Question: "Which is the main topic of the image?"

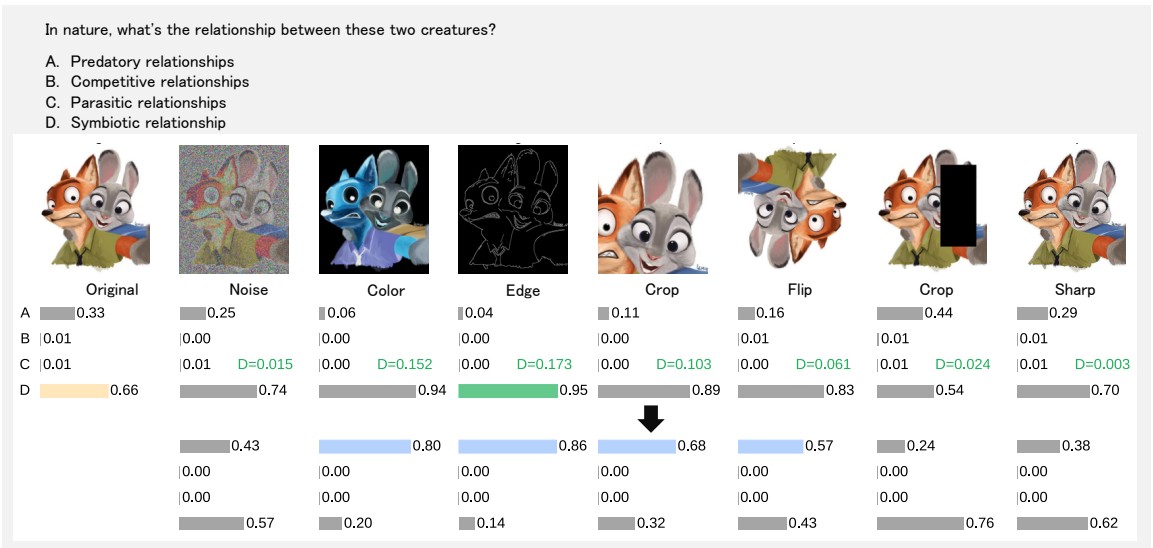

(e) Question: "In nature, what's the relationship between these two creatures?"

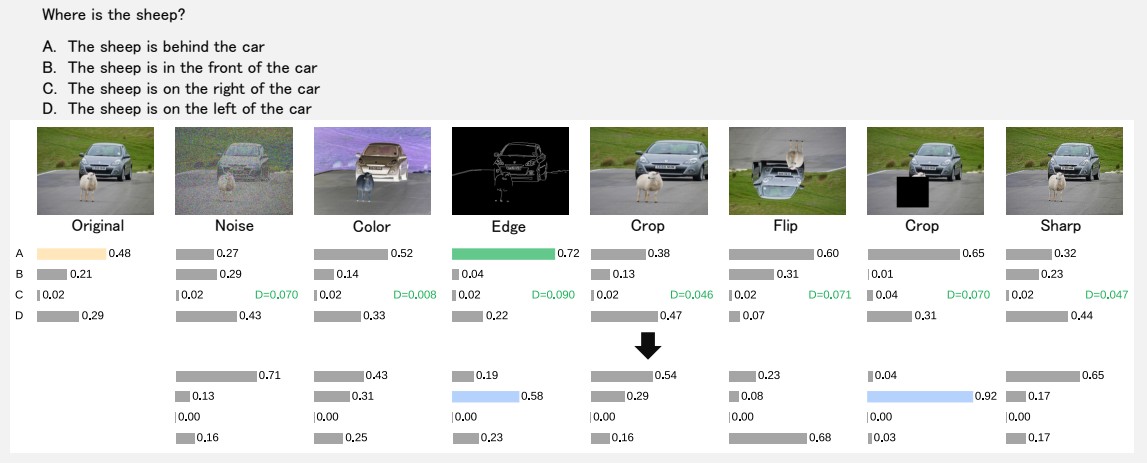

(f) Question: "Where is the sheep?"

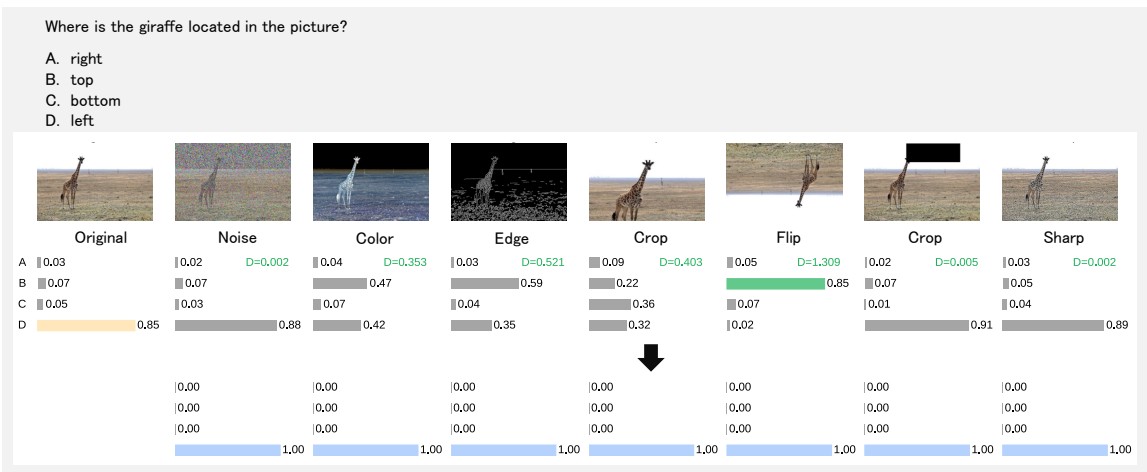

(g) Question: "Where is the giraffe located in the picture?"

Figure 10: Additional case studies utilizing VSCoDe.

