# OpenReview forum: "VSCoDe: Visual-Augmentation Selection for Contrastive Decoding"
_TMLR — Accepted by TMLR_

### Review · Reviewer_PTRk · 2025-05-09

**Summary Of Contributions:**

The authors aim to improve the quality of the responses of Large Vision Language Models at test time with contrastive decoding. Their approach consists of selecting the most appropriate augmentations to apply to the input image to contrast the generation against its unmodified version, reducing the likelihood of decoding incorrect tokens and increasing the chance for the correct ones. For this, they propose the VSCoDe method, which identifies the most suitable augmentations for the problem via a distance metric and applies the augmentations to calibrate the generation, effectively improving the output quality, as demonstrated by extensive experiments on different benchmarks.

**Audience:**

Yes

**Broader Impact Concerns:**

No concerns.

**Claims And Evidence:**

Yes

**Requested Changes:**

The authors should primarily focus on adding missing details regarding the weaknesses highlighted in the above box. Specifically:
- Explain why they apply the distance measure on the normalized logits, potentially showing how performance changes when applied directly on raw outputs.
- Confirm if they tuned the distance metric on the MME benchmark, potentially justifying their decision.
- Explain why the main paper reports a mixture of results between llava-1.5-7b and 13b, potentially uniforming the results using a single version.
- Explain why the authors did not test more recent models, e.g., qwen2-vl, or llava-onevision.
- Explain how the authors set the hparams of their method.
- Demonstrate with an ablation the performance of the coreset strategy with different tau.

**Strengths And Weaknesses:**

Strength:
- The authors propose a training-free approach to improve the quality of the generation of language models with visual capabilities.
- The presentation of the work is curated and detailed, explaining the parts clearly.
- The authors tested different models to confirm the validity of their approach.

Weaknesses:
- Unclear why the authors apply the distance measures directly to the normalized logits (i.e., after softmax). As also confirmed in the Appendix, doing so results in very little change (i.e., at most a difference of 0.04), regardless of the distance metric used (i.e., variation between 0.04 and 0.01, with L1, L2, and L3 performing identically). In addition, it seems like the authors ablated the distance measure on the MME benchmark, one of the benchmarks they use to evaluate their method.
- Unclear why the authors report results mixed between llava-1.5-7b and 13b on the main paper (e.g., Tab 1 is on 7b, Fig 4 is unclear, Tab 2 is on 13b, and Tab 3 is on 7b again). Moreover, it is unclear why the authors present results only for llava 1.5 in the main paper and keep the results for instructblip and qwen in the Appendix. Last, all these model families have more recent versions, e.g., qwen2-vl, or llava-1.6 (also llava-onevision).
- Unclear how the authors set the hparams for their method, i.e., (alpha, beta, T, p, and tau). Also, the work is missing an ablation on the tau hparams, to show how performance changes with different thresholds on the coreset strategy.

---

> ### Author Response · Authors · 2025-07-04
> **Response to reviewer PTRk (1/2)**
>
> Thank you for your constructive feedback to enhance our research. We have considered your suggestions and will adjust our final manuscript accordingly. While we have made every effort to address your comments, we may have unintentionally misunderstood some of concerns. Please feel free to let us know if you have any further questions or need additional clarification.
> ___
> ### **1. Use of normalized logits for measuring distance.**
> We conducted experiments using various types of measurements to find the best method for VSCoDe, including raw logits. Raw logits can differ greatly in scale across images, making them unreliable for direct comparison. By normalizing the logits, we restrict their values to a uniform range and thus stabilize the VSCoDe process by focusing on the relative difference between the distribution of logits. In our experiments, using **raw logits** as the distance metric yielded an MME score of **1346.07**, whereas the **normalized logits** scored **1364.36**. As a result, we found that using distances and softmax logits in our VSCoDe framework significantly helps in choosing the appropriate CD augmentations corresponding to a given query.
>
> |Method | Existence | Count | Position | Color | Posters | Celebrity | Scene | Landmark | Artwork | OCR | Total (± Std) |
> | ---| ---| ---| ---| ---| ---| ---| ---| ---| ---| ---| --- |
> |RAW | 178.0 | 133.0 | 125.66 | 142.66 | 130.74| 135.29| 147.15 | 139.55 | 110.0 | 104.0 | 1346.07 ± 15.70|
> |VSCoDe | 179.00 | 137.33 | 129.67 | 146.67 | 133.95 | 131.88 | 147.40 | 142.00 | 112.50 | 108.50 | **1368.89 ± 20.39** |
>
> Furthermore, in the case of the softmax gain graph shown in Figure 8 (Appendix A), this reflects the additional gain on top of the original model’s performance. Considering that the model already reasonably outputs the given query, it would be difficult to conclude that the overall improvement is marginal. Especially when looking at the average performance across the entire datasets(MME, MMB, VQAv2, POPE, Captioning), using VSCoDe demonstrates higher performance compared to existing methods, and it supports its effectiveness.
> ___
> ### **2. Demonstrating VSCoDe’s generality.**
> In addition to the MME benchmark, which is used mainly to isolate the impact of each augmentation on different question types, we evaluated VSCoDe on VQAv2, MMBench, and POPE for VQA tasks, as well as on MSCOCO for image captioning. The results in Table 3 and Figure 5 confirm that **VSCoDe performs robustly across these diverse tasks and datasets**, indicating that it is not merely tuned to the MME benchmark. Instead, VSCoDe effectively selects the corresponding augmentations that provide effective contrastive decoding for a given query, regardless of the underlying dataset or model conditions.
>
> ___
> ### **3. Incorporating more recent models.**
>
> We would like to clarify that the results for InstructBLIP and Qwen-VL are included in the appendix, as they form part of our broader ablation analysis across diverse models and sizes. Due to page limitations, we were unable to include them in the main. If the page limitation is expanded after the revision, we’ll be able to move these results to the main paper.
>
> Also, following your suggestion, we have run our experiments on more recent model families in the MME dataset, specifically **LLaVA-OneVision** and **Qwen2-VL**. The results for these newer models are summarized below. These results demonstrate that **VSCoDe operates effectively regardless of model types or families** and continues to perform well even with the latest model architectures.
>
>
> | Method       | Augmentation Type | LLaVA-OneVision 7B | Qwen2-VL 7B        |
> |--------------|-------------------|---------------------|---------------------|
> | Vanilla      | –                 | 1441.59 ± 28.29     | 1481.82 ± 8.80      |
> | Single CD    | color             | 1528.28 ± 18.87     | 1513.15 ± 24.88     |
> |              | edge              | 1547.04 ± 34.35     | 1552.14 ± 9.22      |
> |              | sharp             | 1495.46 ± 13.01     | 1516.84 ± 21.19     |
> |              | crop              | 1517.61 ± 16.77     | 1512.10 ± 22.00     |
> |              | erase             | 1509.56 ± 22.29     | 1501.28 ± 21.97     |
> |              | flip              | 1530.90 ± 25.44     | 1541.49 ± 28.76     |
> | VCD          | noise             | 1531.79 ± 29.55     | 1519.61 ± 19.41     |
> | VSCoDe       | all               | **1579.34 ± 15.36**     | 1547.07 ± 25.71     |
> |                      | Coreset               | 1574.25 ± 17.29     | **1552.91 ± 19.82**     |

---

> ### Author Response · Authors · 2025-07-04
> **Response to reviewer PTRk (2/2)**
>
> ### **4. Model-size consistency.**
> We will update all result tables to report performance uniformly using the LLaVA-1.5 7B model. Table 2 will be replaced by the below.
>
> | Method       | Augmentation Type | Existence | Count  | Position | Color  | Posters | Celebrity | Scene  | Landmark | Artwork | OCR    | Total      |
> |--------------|-------------------|-----------|--------|----------|--------|---------|-----------|--------|----------|---------|--------|---------------------|
> | Vanilla      | –                 | 180.00    | 112.00 | 117.67   | 147.00 | 121.02  | 110.24    | 148.80 | 129.15   | 109.35  | 97.00  | 1272.22 ± 28.28     |
> | Single CD    | Color             | 180.00    | 129.33 | 127.33   | 152.67 | 132.45  | 128.06    | 148.30 | 138.35   | 109.75  | 101.00 | 1347.24 ± 21.13     |
> |              | Edge              | 176.00    | 133.00 | 126.67   | 141.00 | 133.61  | 133.35    | 150.40 | 142.45   | 110.70  | 103.50 | 1350.68 ± 6.56      |
> |              | Sharp             | 184.00    | 113.33 | 127.00   | 163.67 | 126.46  | 111.59    | 150.80 | 132.15   | 112.60  | 102.00 | 1323.60 ± 6.51      |
> |              | Crop              | 183.00    | 122.00 | 129.67   | 151.67 | 129.86  | 128.35    | 146.80 | 134.70   | 110.95  | 101.50 | 1338.50 ± 23.83     |
> |              | Erase             | 182.00    | 118.33 | 119.33   | 153.67 | 126.12  | 115.24    | 149.95 | 134.85   | 112.90  | 98.50  | 1310.89 ± 18.82     |
> |              | Flip              | 178.00    | 124.67 | 129.67   | 152.33 | 132.24  | 125.24    | 147.30 | 137.05   | 111.25  | 107.00 | 1344.75 ± 21.05     |
> | VCD          | Noise             | 186.00    | 115.00 | 122.33   | 154.33 | 127.76  | 125.47    | 149.85 | 138.30   | 110.90  | 93.50  | 1323.44 ± 27.36     |
> | Rand CD  | All               | 182.00    | 125.67 | 127.33   | 153.00 | 130.82  | 123.94    | 148.90 | 134.80   | 110.80  | 102.50 | 1339.76 ± 32.19     |
> | VSCoDe       | All               | 179.00    | 137.33 | 129.67   | 146.67 | 133.95  | 131.88    | 147.40 | 142.00   | 112.50  | 108.50 | **1368.89 ± 20.39**     |
> |              | Coreset        | 179.00    | 138.33 | 129.67   | 150.67 | 132.31  | 130.88    | 145.00 | 139.75   | 110.25  | 108.50 | 1364.36 ± 11.50     |
>
> ___
> ### **5. Hyperparameter choices and ablation on τ.**
> Our base implementation details in Section 4.1 follow the settings from VCD (Leng et al., 2023). For more information, we choose α = 1.0 and β = 0.1 for the main experiment. Additionally, we use T = 1.0 and p = 1.0 for the sampling strategy. Furthermore, we provide ablation results on the sampling strategy in Appendix C and implementation details, such as VCD’s noise step, as well as ICD, HALC, and CRG, in Appendix E.
>
> Through your thoughtful suggestion, we conducted an ablation study on acceptance threshold τ to quantify its effect. As described in Section 3.4 (Corset Strategy), a lower τ allows a greater number of augmentations to participate in VSCoDe. In comparison, a higher τ selects only a few augmentations with higher contributions, thereby **reducing the computational cost**. The results are presented in the table below with the LLaVA1.5 7B model. Lower values of τ allow for more augmentations, but despite the increased computational cost, the resulting performance gains remain marginal.
>
> | τ  | # of augs | MME     |
> |-------|-----|---------|
> | –     | 7   | 1368.89 |
> | 0.38  | 6   | 1366.71 |
> | 0.39  | 5   | 1363.20 |
> | 1.13  | 4   | 1364.37 |
> | 1.14  | 3   | 1359.16 |
> | 1.21  | 2   | 1361.63 |
> | 2.68  | 1   | 1350.68 |
>
> Moreover, this **Coreset strategy can lead to performance improvements by removing less effective augmentations** from the set in some time. The table below presents the τ ablation study conducted with InstructBLIP 7B. The results show that, up to a certain τ, the Coreset can outperform the full set of augmentations. In conclusion, depending on the dataset and model, the Coreset strategy can provide gains not only in computational efficiency but also in performance, as demonstrated in Table 3 and Table 7.
>
> | τ  | # of augs | MME     |
> |-------|-----|---------|
> | –     | 7   | 1249.56 |
> | 0.38  | 6   | 1249.56 |
> | 0.58  | 5   | 1254.16 |
> | 0.92  | 4   | 1254.64 |
> | 0.95  | 3   | 1257.23 |
> | 1.38  | 2   | 1249.13 |
> | 2.78  | 1   | 1221.15 |
> Across various datasets and models, **no single CD consistently achieves the highest performance**. However, our **VSCoDe method reliably outperforms** all individual augmentations used, demonstrating stable and superior performance.

---

> > ### Comment · Reviewer_PTRk · 2025-07-21
> >
> > I thank the authors for their clear and complete response. All of my concerns have been addressed.

---

### Review · Reviewer_oq7g · 2025-05-18

**Summary Of Contributions:**

language decoders sometimes produce incorrect outputs, a phenomenon called hallucination.
1. The author aims in digging the impact of different visual augmentation to the visual perception process which is convincing.
2. Based on the motivation/oberservation in line 1, the author propose a novel strategy in VSCoDe that selects contrastive augmentation to empower CD capability without additional training or using external models.
3. The result shows the advancement of the strategy.

**Audience:**

Yes

**Claims And Evidence:**

Yes

**Requested Changes:**

Same as above

**Strengths And Weaknesses:**

## Strength
1. The author propose a Query-Dependent Augmentation Effect section, aiming in demonstrating that various augmentations affect LVLM outputs differently. The result in Table 1 effectively and quantitively shows the impact of various augmentation.

2. The main method is Maximizing Contrast: Selecting Visual Augmentation with the Largest Distance which aims in selecting the visual augmentation that yields the most contrastive outputs relative to inference on the original image. The distance measurement is through measuring distribution difference between the output distributions from original and augmented images.



## Weakness
1. In demostrating various visual augmentations, would the combination of different methods yields a better performance? It seems like that author only focus on invidual augmentation.
2. In exploring the distance measurement for the method design, the author select through measureing the output distribution. However, the more intuitive way is through the semantic distance (such as using the CLIP visual encoder to do the distance selection). This distance measurement should be considered.
3. Based on the motivation experiments and main experiments, the most effective visual augmentation normally has a connection with the question. If so, directly using the question-augmentation matching such as using text encoder would directly select the most effective augmentation?

---

> ### Author Response · Authors · 2025-07-04
> **Response to reviewer oq7g (1/2)**
>
> Thank you for your valuable and insightful feedback, which aims to improve our research paper. We have considered your suggestions and will revise our final manuscript accordingly. Although we have endeavored to address all your comments, there may have been some misunderstandings on our part. Therefore, please do not hesitate to reach out if you have any further questions or require additional clarification.
> ___
> ### **1. Combination of augmentations.**
>
> While combining multiple augmentations could improve performance, our primary goal is to demonstrate the importance of selecting contrastive augmentation and suggest a strategy for choosing the augmentation. Based on your feedback, we conducted **experiments with various pairs of augmentations** on the MME benchmark using the LLaVA-1.5 7B model and report the results in the table below.
>
> |Method | Existence | Count | Position | Color | Posters | Celebrity | Scene | Landmark | Artwork | OCR | Total|
> | ---| ---| ---| ---| ---| ---| ---| ---| ---| ---| ---| --- |
> |Edge+Crop | 184.0 | 130.67 | 120.0 | 132.00 | 143.13 | 130.35 | 153.25 | 143.05 | 105.6 | 109.5 | 1351.55 |
> |Color+Sharp | 178.0 | 129.67 | 126.0 | 152.67 | 131.77 | 128.24 | 150.15 | 139.15 | 109.9 | 105.5 | 1351.04 |
> |Crop+Flip | 180.0 | 121.33333333333334 | 127.0 | 144.0 | 135.71 | 130.82 | 150.4 | 141.45 | 113.4 | 108.0 | 1352.12 |
> |VSCoDe-comb | 181.0 | 126.33 | 128.33 | 132.67 | 139.25 | 132.59 | 151.6 | 143.55 | 105.7 | 116.0 | 1357.02 |
> |VSCoDe (all) | 179.00 | 137.33 | 129.67 | 146.67 | 133.95 | 131.88 | 147.40 | 142.00 | 112.50 | 108.50 | **1368.89** |
>
> Although combining multiple augmentations can yield higher overall MME scores than single augmentations. Furthermore, leveraging such combined augmentations as candidates in VSCoDe can also result in better performance than using them individually. However, our results show that **VSCoDe (all)** - which utilizes a complete set of 7 single augmentations - achieves even higher performance than the **VSCoDe-comb with these three combined combinations**. For example, in the color category, using single 'Color' augmentation achieves better performance and contributes to the superior performance of VSCoDe (all). This demonstrates that simply relying on strong performance augmentations is not sufficient; rather, **constructing a complementary augmentation set that aligns well with the given query is more critical** for contrastive decoding. Here, VSCoDe can maximize this contrastive effect by automatically selecting the most suitable augmentation for each query from the augmentation candidates.
> ___
> ### **2. Limitations of semantic encoders for contrastiveness.**
>
> Thank you for the suggestion. While a semantic vision encoder (e.g., CLIP) can measure overall semantic differences between images, they do not capture the differences relevant to answering a given query. In other words, maximizing semantic distance from the original image does not guarantee contrast with respect to the query. For the experiment, we computed the distance between CLIP embeddings of the original and augmented images, selecting the augmentation with the largest semantic shift. We show experimental results on the MME benchmark with LLaVA-1.5 7B, demonstrating this distinction in the table below. VSCoDe (all) records better in most categories and total scores, underscoring the importance of considering the semantics of the query.
>
> |Method | Existence | Count | Position | Color | Posters | Celebrity | Scene | Landmark | Artwork | OCR | Total|
> | ---| ---| ---| ---| ---| ---| ---| ---| ---| ---| ---| --- |
> |VSCoDe-CLIP | 177.00 | 128.66| 123.33| 145.0 | 136.39 | 128.58 | 149.2 | 140.7 | 111.35 | 98.0 | 1338.23 |
> |VSCoDe (all) | 179.00 | 137.33 | 129.67 | 146.67 | 133.95 | 131.88 | 147.40 | 142.00 | 112.50 | 108.50 | **1368.89** |

---

> ### Author Response · Authors · 2025-07-04
> **Response to reviewer oq7g (2/2)**
>
> ___
> ### **3. About utilizing text encoders.**
>
> Thank you for the thoughtful suggestion. As mentioned in our paper, it is possible to use a text encoder since the contrastive augmentation is related to the semantics of each query. However, **text encoder-based approaches require additional modules or training** to classify or determine the contrastive augmentation based on the given query text, while VSCoDe is training-free. Furthermore, our experimental results demonstrate that **no single augmentation consistently dominates across a query type**. This highlights the importance of considering not only the text query but also the visual content of the image when selecting the most appropriate augmentation. VSCoDe can automatically select the augmentation that provides the **contrast by considering both the image and text query during the CD**.
>
> Furthermore, our VSCoDe method is also compatible with **incorporating text-based augmentations as part of the candidate set**. The table below presents results when **ICD**[1] is included alongside our original augmentation set. Since VSCoDe selects the augmentation with the high contrast among the given candidates, expanding the candidate pool to include diverse types of augmentations can potentially lead to further performance improvements.
>
> | Method           | Augmentation Type | Existence | Count  | Position | Color  | Posters | Celebrity | Scene  | Landmark | Artwork | OCR    | Total           |
> |------------------|-------------------|-----------|--------|----------|--------|---------|-----------|--------|----------|---------|--------|------------------|
> | Vanilla          | –                 | 180.00    | 112.00 | 117.67   | 147.00 | 121.02  | 110.24    | 148.80 | 129.15   | 109.35  | 97.00  | 1272.22          |
> | VCD              | Noise             | 186.00    | 115.00 | 122.33   | 154.33 | 127.76  | 125.47    | 149.85 | 138.30   | 110.90  | 93.50  | 1323.44  |
> | ICD              | Text              | 183.00    | 122.67 | 120.33   | 165.33 | 128.16  | 112.82    | 150.65 | 130.90   | 113.10  | 98.50  | 1325.47          |
> | VSCoDe           | All               | 179.00    | 137.33 | 129.67   | 146.67 | 133.95  | 131.88    | 147.40 | 142.00   | 112.50  | 108.50 | 1368.89          |
> | VSCoDe + ICD     | All + ICD         | 179.00    | 136.67 | 126.33   | 149.00 | 137.14  | 130.41    | 148.10 | 138.85   | 110.40  | 114.50 | **1370.40**          |
>
> ___
> [1] Mitigating Hallucinations in Large Vision-Language Models with Instruction Contrastive Decoding

---

### Review · Reviewer_f7iM · 2025-06-11

**Summary Of Contributions:**

This paper proposes an adaptive method for selecting augmented images for contrastive decoding(CD) across a range of vision-language tasks. By identifying and utilizing the most effective visual augmentations, the approach enhances the performance of large vision-language models(LVLMs) without additional training or reliance on external models.

**Audience:**

Yes

**Claims And Evidence:**

Yes

**Requested Changes:**

Apply augmentation strategies with certain randomness multiple times to enhance stability—for example, perform cropping or erasing several times and average the results. For flipping, you can also try using different rotation angles.(Suggestions for Improving the quality)

**Strengths And Weaknesses:**

Strengths:
This work adaptively selects task-specific visual augmentations to improve the effectiveness of contrastive decoding. By applying simple image manipulations, it effectively corrects the output of large vision-language models without requiring additional model modifications.
Weaknesses:
1. Limited novelty: The proposed method appears to be a combination of existing techniques， which applies known visual augmentations within a contrastive decoding framework. This integration may limit the originality of the contribution.
2. Potential Limitations of Augmentation: In some cases, visual augmentations for contrastive decoding seem to negatively impact model performance. For example, in Table 1, all augmentation strategies appear to degrade performance on OCR tasks. Similarly, in Table 2, performance on Scene tasks is worse than that of Vanilla decoding method.
3. Randomness in augmentations: Some augmentations, such as cropping, may randomly retain the entire task-relevant objects, which is not necessarily beneficial for contrastive decoding. In addition, flipping can reinforce symmetrical location biases in position-sensitive tasks, which are not necessarily severe hallucinations in the original output.

---

> ### Author Response · Authors · 2025-07-04
> **Response to reviewer f7iM (1/2)**
>
> First of all, thank you for your constructive and thoughtful feedback to improve our research paper. We have taken your feedback into account and will prepare our final manuscript accordingly. While we have to make every effort to address your comments, we may have inadvertently misunderstood some of your concerns. Therefore, please do not hesitate to let us know if you have any further questions or require additional clarification.
>
> ___
> ### **1. Novelty of VSCoDe.**
> While prior works on contrastive decoding apply a single, fixed augmentation type [1], [2], [3], we suggest an adaptive, query-dependent augmentation selection strategy. We demonstrate the need for using query-dependent augmentation for each query and suggest a distance metric to choose the contrastive augmentation. Unlike fixed augmentation-based methods, which may fail for certain types of queries, **VSCoDe selects contrastive augmentation adaptively**, resulting in strong performance across different categories, as shown in Table 2.
>
> The key strength of VSCoDe’s automatic contrastive augmentation selection lies in its ability to utilize a diverse set of augmentation candidates, including those that may not be considered in fixed single-augmentation approaches. This flexibility is not limited to either the image or text modality alone; instead, **VSCoDe jointly considers both the image and the textual query to automatically select the augmentation** that yields the most substantial contrastive effect.
>
> Furthermore, our VSCoDe method is designed not only to work with visual augmentations but also to incorporate text-based augmentations into the candidate set. The table below presents results when ICD is added to our original augmentation set. By evaluating all candidates—regardless of whether they originate from the visual or textual modality—**VSCoDe effectively identifies the augmentation that provides the strongest contrastive signal for a given query**. This demonstrates that VSCoDe is not restricted to a specific modality and highlights its flexibility in leveraging a diverse set of augmentations to enhance performance further.
>
> | Method           | Augmentation Type | Existence | Count  | Position | Color  | Posters | Celebrity | Scene  | Landmark | Artwork | OCR    | Total           |
> |------------------|-------------------|-----------|--------|----------|--------|---------|-----------|--------|----------|---------|--------|------------------|
> | Vanilla          | –                 | 180.00    | 112.00 | 117.67   | 147.00 | 121.02  | 110.24    | 148.80 | 129.15   | 109.35  | 97.00  | 1272.22          |
> | ICD              | Text              | 183.00    | 122.67 | 120.33   | 165.33 | 128.16  | 112.82    | 150.65 | 130.90   | 113.10  | 98.50  | 1325.47          |
> | VSCoDe           | All               | 179.00    | 137.33 | 129.67   | 146.67 | 133.95  | 131.88    | 147.40 | 142.00   | 112.50  | 108.50 | 1368.89          |
> | VSCoDe + ICD     | All + ICD         | 179.00    | 136.67 | 126.33   | 149.00 | 137.14  | 130.41    | 148.10 | 138.85   | 110.40  | 114.50 | **1370.40**          |
>
> ___
> ### **2. Building a complete augmentation set.**
> As noted in the conclusion section of the paper, the effectiveness of VSCoDe depends on the candidate pool containing at least one truly contrastive augmentation for each query. Consequently, it may underperform in categories where this assumption fails. As mentioned, VSCoDe gets a low score in the scene category, as shown in Table 2. In this case, combining crop and edge augmentations makes a higher score. The table below shows the scores of VSCoDe using the 3 combinations of augmentations (VSCoDe-comb) and our original VSCoDe with 7 single augmentations (VSCoDe (all)). Using a combination of edge and crop yields higher performance in VSCoDe, but the overall score is higher in VSCoDe (all). It is important to find the complement set of augmentations, and we leave identifying an optimal augmentation set for future work.
>
> |Method | Existence | Count | Position | Color | Posters | Celebrity | Scene | Landmark | Artwork | OCR | Total|
> | ---| ---| ---| ---| ---| ---| ---| ---| ---| ---| ---| --- |
> |Vanilla| 180.00    | 112.00  | 117.67   | 147.00  | 121.02  | 110.24    | 148.80  | 129.15   | 109.35  | 97.00  | 1272.22    |
> |Edge+Crop | 184.0 | 130.67 | 120.0 | 132.00 | 143.13 | 130.35 | **153.25** | 143.05 | 105.6 | 109.5 | 1351.55 |
> |Color+Sharp | 178.0 | 129.67 | 126.0 | 152.67 | 131.77 | 128.24 | 150.15 | 139.15 | 109.9 | 105.5 | 1351.04 |
> |Crop+Flip | 180.0 | 121.33333333333334 | 127.0 | 144.0 | 135.71 | 130.82 | 150.4 | 141.45 | 113.4 | 108.0 | 1352.12 |
> |VSCoDe-comb | 181.0 | 126.33 | 128.33 | 132.67 | 139.25 | 132.59 | **151.60** | 143.55 | 105.7 | 116.0 | 1357.02 |
> |VSCoDe (all) | 179.00 | 137.33 | 129.67 | 146.67 | 133.95 | 131.88 | 147.40 | 142.00 | 112.50 | 108.50 | **1368.89** |

---

> ### Author Response · Authors · 2025-07-04
> **Response to reviewer f7iM (2/2)**
>
> ___
> ### **3. Balancing randomness and cost.**
> As you mentioned, random augmentations may have some possibilities for retaining the image’s task-relevant information. On the other hand, utilizing an external semantic model (e.g., object detection) could more reliably target query-relevant regions; however, it requires additional cost, as shown in Table 3. To overcome the unstable randomness without relying on external models, we use various augmented images and then select the contrastive image. This strategy enables **robust performance improvements while incurring only a marginal additional computational cost without relying on any external models**.
>
> ___
> ### **4. Minimizing randomness.**
> We provide the results for averaged random augmentations in the table below. We measure the distance for crop and erase augmentations by averaging over five independent random trials (VSCoDe-Mean). While it shows robust scores with low standard deviation, its overall score is lower than VSCoDe (all). Since averaging the distances mixes contrastive and retained images together, it is hard to expect high contrast on these averaged augmentations.
>
> | Method | MME Score |
> | ---| ---|
> | VSCoDe-Mean 	 | 1345.42 ± 5.03 |
> | VSCoDe (all)		 | 1368.89 ± 20.39 |
>
> We also provide the MME score for different angles of rotation augmentation. Flipping provides shifts on both left-right and top-bottom, generating strong contrast, resulting in a high MME score along with 135 degree rotation. **In general, providing properly aligned higher visual contrast can lead to an improvement in the CD effect**.
>
> | Augmentation | Strength   | MME Score           |
> |--------------|------------|---------------------|
> | Rotate       | 45°        | 1306.46 ± 18.45     |
> |              | 90°        | 1316.22 ± 25.06     |
> |              | 135°       | 1345.60 ± 11.10     |
> |              | Flip (180°)| 1344.75 ± 7.00      |
>
>
>
> ___
> [1] Mitigating Hallucinations in Large Vision-Language Models with Instruction Contrastive Decoding
>
> [2] HALC: Object Hallucination Reduction via Adaptive Focal-Contrast Decoding
>
> [3] Contrastive Region Guidance: Improving Grounding in Vision-Language Models without Training

---

> > ### Comment · Reviewer_f7iM · 2025-07-21
> >
> > Most of my concerns are addressed and thanks for your effort.

---

### Author Response · Authors · 2025-07-04
**General Response**

**Dear reviewers (@PTRk, @oq7g, @f7iM),**

We sincerely thank you for your considerate reviews and insightful feedback. Your thoughtful comments have provided us with a valuable opportunity to improve our work further and enhance its contribution to academia. We are deeply encouraged by your recognition of the contributions, significance, and strengths of our research, and it has been a strong source of motivation for us.
___
### Key Strengths of VSCoDe Acknowledged by Reviewers

- Contribution to **adaptive selection method** of task-specific augmentations to improve contrastive decoding (@f7iM, @oq7g)

- **Training-free approach** to improve LVLM generation quality **without additional models and changes**. (@PTRk, @f7iM)

- **Extensive validation** across multiple models and effectively and quantitatively shows the impact. (@oq7g, @PTRk)

- **Clear and well-organized** presentation. (@PTRk)
___
### Core Contributions of Our Work

**Analyzing the contrastive impact of visual augmentations on each query in LVLMs**:
We conducted an empirical analysis to examine how different visual augmentations interact with various types of queries in large vision-language models (LVLMs). Our findings reveal that **leveraging the proper contrastive information** for a given query can significantly improve model performance. Based on the valuable feedback from Reviewers @f7iM and @oq7g, we extended our analysis to include results from **combinations of multiple augmentations**, further validating the importance of choosing augmentations adaptively.


**Automatic selection of query-dependent visual augmentations without extra modules**:
Our proposed method, VSCoDe, **automatically selects** contrastive augmentation adaptively for a given query **without requiring additional modules and training**. It leverages a simple but effective distance-based measurement, which aligns well with relevant augmentation categories and consistently enhances performance. In response to the insightful comments from Reviewers @PTRk and @oq7g, we conducted **additional ablation studies using different measurement strategies**—including raw logits and CLIP-based metrics—to provide a more comprehensive understanding of how the selection mechanism functions.


**Robust empirical validation and superior performance**:
We validated the effectiveness of VSCoDe across a range of benchmark datasets, including MME, MMBench, VQAv2, POPE, and MSCOCO Captioning. These results demonstrate that our method consistently improves LVLM performance across various tasks. Following the helpful suggestion from Reviewer @PTRk, we further strengthened our empirical validation by including **ablation studies on the acceptance threshold τ** and by evaluating VSCoDe on recent state-of-the-art LVLMs such as **LLaVA-OneVision** and **Qwen2-VL**. These experiments confirm that VSCoDe maintains its effectiveness even with the latest model architectures.
___
Furthermore, we make efforts to address the questions and concerns raised by reviewers individually by adding our additional experiments and clarifications. We hope that these comments and contributions resolve any remaining questions, provide a clearer understanding of our work, and support its improvement and development.

Sincerely,

Authors

---

### Decision · Action_Editor_8UC3 · 2025-08-16

**Recommendation:** Accept as is

**Audience:**

Yes

**Audience Explanation:**

Hallucination is an important problem that is faced by VLMs which limits their reliability. Hallucination mitigation is an important research topic, the method is simple, it considers a set of image augmentations for the contrastive decoding and dynamically selects the most suitable augmentation for the test image based on augmentation-query distance. The is training free and shown effective.

**Claims And Evidence:**

Yes

**Claims Explanation:**

The paper shows improved performance and reduced hallucinations in various and strong experiments justifying the effectiveness of the proposed method.